# FLASHVID: EFFICIENT VIDEO LARGE LANGUAGE MODELS VIA TRAINING-FREE TREE-BASED SPATIOTEMPORAL TOKEN MERGING

**Ziyang Fan**[1]  **Keyu Chen**[1]  **Ruilong Xing**[1]  **Yulin Li**[1]  **Li Jiang**[2,3]  **Zhuotao Tian**[1,3*]
[1]Harbin Institute of Technology, Shenzhen  [2]The Chinese University of Hong Kong, Shenzhen
[3]Shenzhen Loop Area Institute

## ABSTRACT

Although Video Large Language Models (VLLMs) have shown remarkable capabilities in video understanding, they are required to process high volumes of visual tokens, causing significant computational inefficiency. Existing VLLMs acceleration frameworks usually compress spatial and temporal redundancy independently, which overlooks the spatiotemporal relationships, thereby leading to suboptimal spatiotemporal compression. The highly correlated visual features are likely to change in spatial position, scale, orientation, and other attributes over time due to the dynamic nature of video. Building on this insight, we introduce FlashVID, a training-free inference acceleration framework for VLLMs. Specifically, FlashVID utilizes Attention and Diversity-based Token Selection (ADTS) to select the most representative tokens for basic video representation, then applies Tree-based Spatiotemporal Token Merging (TSTM) for fine-grained spatiotemporal redundancy elimination. Extensive experiments conducted on three representative VLLMs across five video understanding benchmarks demonstrate the effectiveness and generalization of our method. Notably, by retaining only **10%** of visual tokens, FlashVID preserves **99.1%** of the performance of LLaVA-OneVision. Consequently, FlashVID can serve as a training-free and plug-and-play module for extending long video frames, which enables a **10×** increase in video frame input to Qwen2.5-VL, resulting in a relative improvement of **8.6%** within the same computational budget. Code is available at https://github.com/Fanziyang-v/FlashVID.

# 1 INTRODUCTION

Recent advances in Video Large Language Models (VLLMs) (Li et al., 2025a; Zhang et al., 2024; Bai et al., 2025b; Comanici et al., 2025) have demonstrated promising capabilities in video understanding tasks. However, processing large numbers of visual tokens incurs substantial computational and memory overhead, as the attention mechanism scales quadratically with sequence length, limiting practical deployment. To address this challenge, visual token compression (Chen et al., 2024; Yang et al., 2025c; Zhang et al., 2025e) has emerged as a promising approach, leveraging the inherent redundancy in visual inputs to reduce sequence length by removing or merging less informative tokens, thereby enabling efficient inference without significant performance degradation.

While advances have been achieved in visual token compression for images (Bolya et al., 2023; Chen et al., 2024; Yang et al., 2025c; Zhang et al., 2025e), extending these methods to video remains largely underexplored. Videos inherently exhibit both *spatial* redundancy within frames and *temporal* redundancy across frames, rendering frame-wise compression strategies suboptimal due to their neglect of temporal dynamics and correlations. This gap highlights the need for compression techniques specifically designed for the spatiotemporal structure of video inputs in VLLMs.

**Motivation.** Recent VLLM acceleration methods (Huang et al., 2025; Shen et al., 2025; Shao et al., 2025a) typically adopt a three-stage pipeline: (1) *video partition*, grouping consecutive frames

---

*Corresponding author: Zhuotao Tian (tianzhuotao@hit.edu.cn)

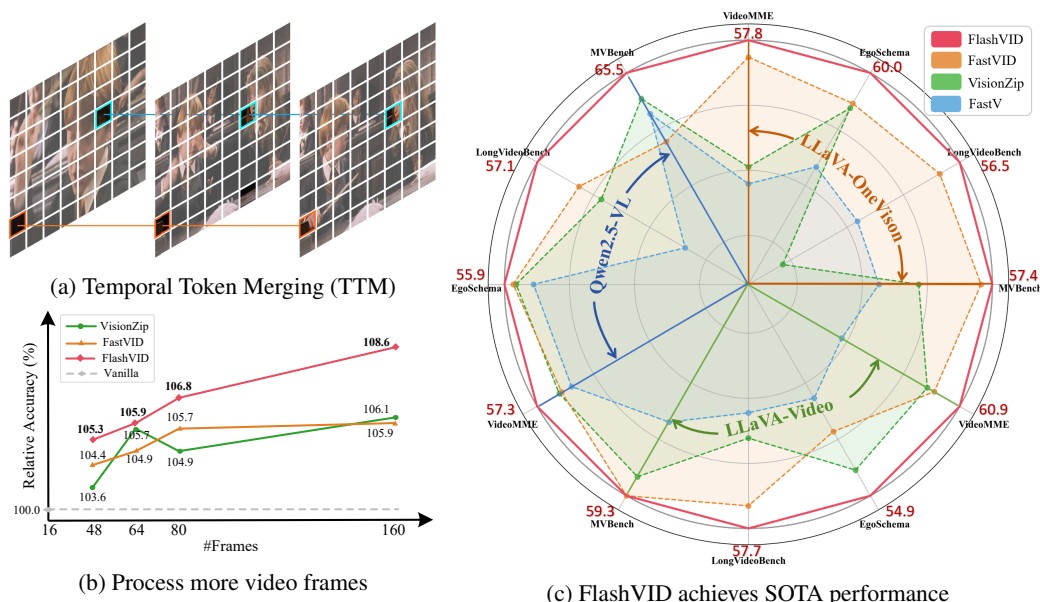

(a) Temporal Token Merging (TTM)

(b) Process more video frames

(c) FlashVID achieves SOTA performance

Figure 1: **Performance of FlashVID.** (a) TTM may merge less correlated visual tokens, failing to capture fine-grained video dynamics. (b) FlashVID can enable Qwen2.5-VL to process **10×** video frames, significantly improving the relative performance by **8.6%** while maintaining overall computational budget. (c) FlashVID significantly outperforms current SOTA acceleration frameworks (e.g., FastV, VisionZip, FastVID) on three representative VLLMs.

with similar semantics to avoid information mixing; (2) *frame-wise token selection*, identifying informative tokens—often guided by [CLS] attention—for basic video representation; and (3) *spatiotemporal compression*, further reducing redundancy at the segment level.

However, previous methods typically compress temporal and spatial redundancy independently. Such decoupled strategies overlook the intrinsic spatiotemporal relationships in videos. Moreover, temporal redundancy is commonly defined as the consistency of visual features at fixed spatial locations across consecutive frames. Due to the dynamic nature of video, the most semantically similar visual elements are likely to experience changes in spatial position, scale, orientation, and other attributes over time. Consequently, the most correlated visual features in adjacent frames may not reside at the same spatial location. As depicted in Fig. 1a, the Temporal Token Merging (TTM) strategy fails to capture video dynamics, erroneously merging less correlated tokens and distorting the video representations. Relying on such a rigid spatial correspondence for temporal redundancy compression may introduce noise, further misleading the model. So, a natural question arises: *"How can we achieve a decent spatiotemporal compression by jointly modeling spatial and temporal redundancy, while accounting for the dynamic characteristics of video?"*

**Our Solution.** To address this challenge, we introduce **FlashVID**, a novel training-free acceleration framework for VLLMs that effectively reduces spatiotemporal redundancy while preserving critical visual content. Specifically, at the core of FlashVID is the **T**ree-based **S**patiotemporal **T**oken **M**erging (TSTM) mechanism, which explicitly models both spatial and temporal redundancy through hierarchical spatiotemporal redundancy trees. TSTM enables structured token merging across frames and within frames, allowing for joint spatiotemporal compression that respects the natural structure of video data. However, directly constructing spatiotemporal trees based on the raw video features with excessive noise and redundancy may not focus on the most representative visual information in each frame, or even be biased towards the major but unimportant visual information, thereby affecting the final performance. To alleviate this issue, we further introduce the **A**ttention and **D**iversity-based **T**oken **S**election (ADTS) module, which prioritizes representative tokens in each frame. To this end, through the initial filtering of informative tokens using ADTS and subsequent merging via TSTM, FlashVID accomplishes efficient compression that adjusts to the dynamic attributes of video content while preserving crucial semantics.

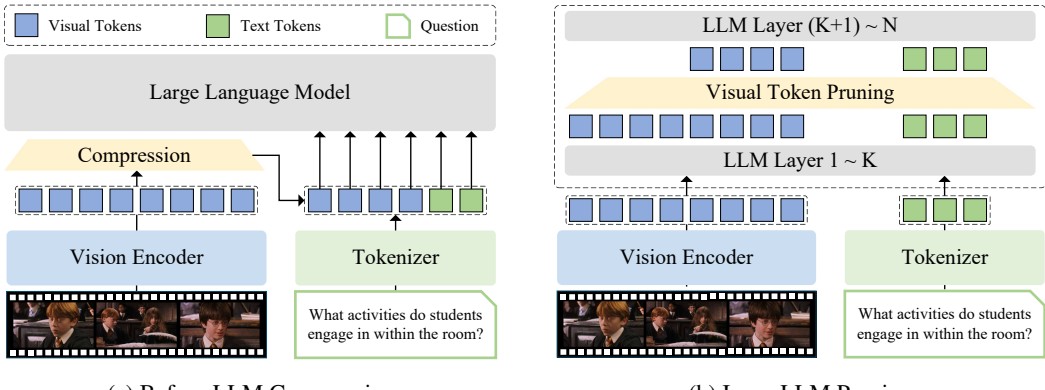

(a) Before-LLM Compression          (b) Inner-LLM Pruning

Figure 2: **Efficient inference paradigms.** State-of-the-art acceleration frameworks can be mainly divided into three categories: 1) Before-LLM Compression; 2) Inner-LLM Pruning; and 3) Hybrid Compression, where the hybrid compression can be viewed as a trade-off of the Before-LLM Compression and Inner-LLM Pruning strategy.

Extensive experiments have been conducted on five video understanding benchmarks (Fu et al., 2025a; Mangalam et al., 2023; Wu et al., 2024; Li et al., 2024a; Zhou et al., 2025) and three representative VLLMs, *i.e.*, LLaVA-OneVision (Li et al., 2025a), LLaVA-Video (Zhang et al., 2024), and Qwen2.5-VL (Bai et al., 2025b). As shown in Fig. 1b and Fig. 1c, FlashVID outperforms previous state-of-the-art methods by a large margin across all settings. Notably, FlashVID achieves **99.1%** relative accuracy to vanilla LLaVA-OneVision while pruning **90%** visual tokens. When integrated into Qwen2.5-VL, FlashVID enables processing of up to $10\times$ more video frames, yielding an **8.6%** performance gain over the vanilla model with 16 sampled frames under the same computational budget, highlighting its ability to unlock longer temporal context for better video understanding.

To summarize, our main contributions are threefold:

- In this work, we identify that existing token compression methods fail to effectively model the dynamic and evolving nature of video content, leading to suboptimal performance.
- We propose FlashVID, a training-free VLLMs acceleration method that introduces the Tree-based Spatiotemporal Token Merging (TSTM) to jointly model spatial and temporal redundancy across frames, complemented by Attention and Diversity-based Token Selection (ADTS) to obtain the semantically representative content within each frame.
- Extensive experiments show that FlashVID improves inference efficiency with negligible performance drop, and enables the use of longer input sequences for better video understanding within the constrained computational budget.

## 2 BACKGROUND AND MOTIVATION

In this section, we provide a brief overview of the underlying concepts in this study in Sec. 2.1, and highlight the key observations in Sec. 2.2, which offer valuable insights for our approach.

### 2.1 PRELIMINARIES

**VLLMs inference pipeline.** The inference of VLLMs consists of three stages: **(1) Encoding.** A vision encoder (e.g., CLIP (Radford et al., 2021) and SigLIP (Zhai et al., 2023)) processes each frame independently, producing $N_v$ visual embeddings per frame, which are projected into text space via a modality connector to form $H_v \in \mathbb{R}^{F \times N_v \times d}$. Text queries are embedded as $H_t \in \mathbb{R}^{N_t \times d}$. **(2) Prefilling.** Each LLM layer $l$ computes self-attention over $H$ via:

$$\mathbf{Q}^l = H^l \mathbf{W}_Q^l, \quad \mathbf{K}^l = H^l \mathbf{W}_K^l, \quad \mathbf{V}^l = H^l \mathbf{W}_V^l, \tag{1}$$

with $\mathbf{W}_Q^l, \mathbf{W}_K^l, \mathbf{W}_V^l \in \mathbb{R}^{d \times d}$. The key-value pairs are stored in the KV Cache for decoding acceleration. **(3) Decoding.** Response tokens are generated auto-regressively. At step $t$, only the new

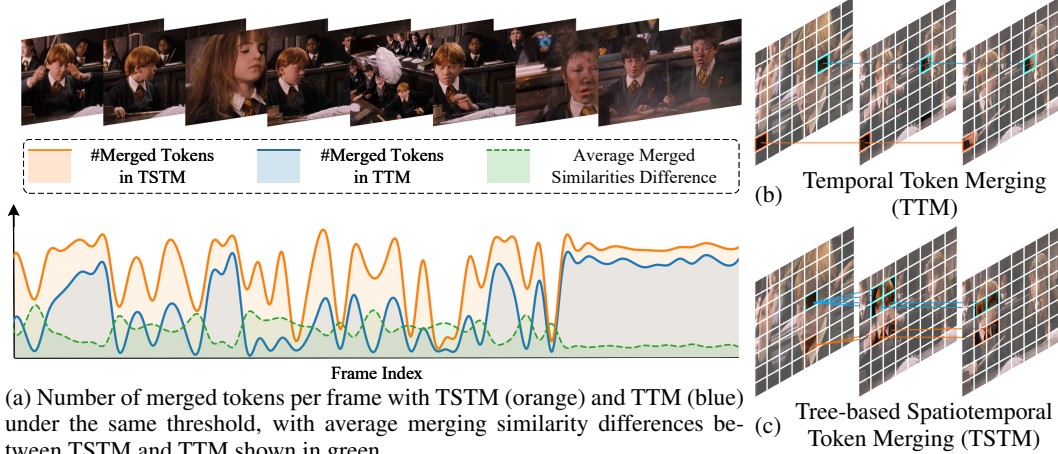

(a) Number of merged tokens per frame with TSTM (orange) and TTM (blue) under the same threshold, with average merging similarity differences between TSTM and TTM shown in green.

(b) Temporal Token Merging (TTM)

(c) Tree-based Spatiotemporal Token Merging (TSTM)

Figure 3: **Comparison of spatiotemporal redundancy compression.** (a) TSTM merges more tokens than TTM under the same threshold and achieves higher inter-frame merging similarity by flexibly capturing fine-grained video dynamics. (b) TTM enforces rigid spatial correspondences, often overlooking dynamic variations in videos and merging less correlated visual tokens. (c) TSTM models video redundancy via spatiotemporal redundancy trees, capturing fine-grained spatiotemporal relationships. More visualizations are provided in Appendix E.

token $h_t$ is projected to $(\mathbf{K}_t, \mathbf{V}_t)$, which update the cache:

$$\mathbf{K} \leftarrow \text{concat}[\mathbf{K}, \mathbf{K}_t], \quad \mathbf{V} \leftarrow \text{concat}[\mathbf{V}, \mathbf{V}_t]. \quad (2)$$

Such a caching mechanism substantially improves decoding efficiency.

**Efficiency bottleneck analysis.** While VLLMs have achieved remarkable performance on video understanding tasks, their efficiency remains a key challenge due to the heavy computational and memory overhead when processing a large number of visual tokens. Most of this cost stems from the LLM backbone, where the self-attention mechanism and Feed-Forward Networks (FFNs) dominate the computational complexity. Given a model with $L$ Transformer layers, the total Floating Point Operations (FLOPs) can be formulated as:

$$\text{FLOPs} = L \times (4nd^2 + 2n^2d + 2ndm), \quad (3)$$

with $n$ denoting the sequence length, $d$ the hidden dimension, and $m$ the intermediate dimension of FFNs. In video understanding, the number of visual tokens $n_v$ dominates the sequence length $n$, typically exceeding textual tokens $n_t$ by orders of magnitude. This imbalance underscores the necessity to compress visual tokens for efficient inference in VLLMs.

**Efficient inference paradigms.** As illustrated in Fig. 2, visual token compression frameworks can be grouped into three paradigms: *Before-LLM*, *Inner-LLM*, and *Hybrid Compression*. Compressing tokens only inside the LLM is inefficient, as all visual tokens must still be processed in the shallow layers; thus, reducing tokens before the LLM is critical for reducing overhead. Existing methods (Yang et al., 2025c; Shen et al., 2025) adopt single-stage compression before the LLM, but extreme compression risks losing important visual information. Hybrid compression provides a balance: it retains sufficient tokens as LLM input while further pruning within the LLM to meet computational budget. Training-based approaches (Zhang et al., 2025d; Cai et al., 2025; Hu et al., 2024; Shao et al., 2025b) can mitigate this inefficiency but demand substantial computing resources; in this work, we focus on training-free strategies. A more comprehensive review is provided in Appendix D.

## 2.2 KEY OBSERVATIONS

We summarize two key observations about spatiotemporal redundancy in videos: (1) *Temporal redundancy is not bound to fixed spatial locations.* Semantically consistent elements in videos often

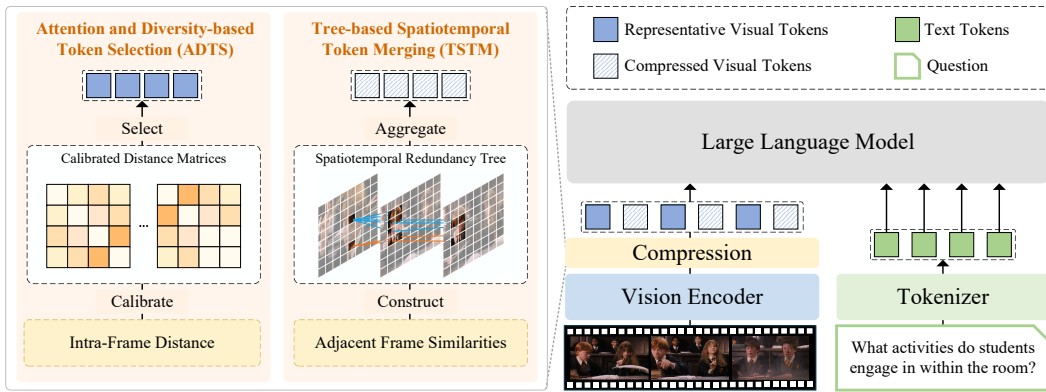

Figure 4: **Overview of our FlashVID.** FlashVID compresses visual tokens by two synergistic modules: (1) ADTS prioritizes spatiotemporally informative tokens while ensuring feature diversity by solving a calibrated Max-Min Diversity Problem (MMDP); (2) TSTM models redundancy by spatiotemporal redundancy trees, which effectively capture fine-grained video dynamics.

shift in spatial position, scale, or appearance due to motion and scene dynamics, making rigid spatial correspondence across frames unreliable (Huang et al., 2025). (2) *Spatial and temporal redundancy are inherently coupled.* Redundant regions within a single frame frequently persist across multiple frames. Decoupled spatiotemporal redundancy compression overlooks the intrinsic spatiotemporal relationships, leading to suboptimal compression.

These insights suggest that existing frameworks lack a unified, structure-aware mechanism to capture spatiotemporal relationships under dynamic video conditions, which motivates our hierarchical tree-based spatiotemporal redundancy compression. A conceptual comparison with prior approaches is illustrated in Fig. 3, highlighting the unique advantages of our design.

## 3 METHODOLOGY

### 3.1 OVERVIEW

As illustrated in Fig. 4, FlashVID integrates two synergistic modules: 1) Attention and Diversity-based Token Selection (ADTS), which first selects informative and diverse tokens for robust video representations; 2) Tree-based Spatiotemporal Token Merging (TSTM), which further minimizes spatiotemporal redundancy while preserving critical visual information.

### 3.2 TREE-BASED SPATIOTEMPORAL TOKEN MERGING

Videos exhibit dynamic variations in spatial position, scale, and appearance, posing challenges for spatiotemporal redundancy compression. To alleviate this, we propose Tree-based Spatiotemporal Token Merging (TSTM), which models the video redundancy via spatiotemporal redundancy trees.

**Construct spatiotemporal redundancy trees.** Given video features $E_v \in \mathbb{R}^{F \times N_v \times d}$, TSTM progressively builds spatiotemporal redundancy trees. First, we compute the cosine similarity matrix between visual features in adjacent frames:

$$S^{(f)} = \cos(E_v^{(f)}, E_v^{(f+1)}) \in \mathbb{R}^{N_v \times N_v}, \tag{4}$$

where $S^{(f)}(j, k)$ measures the feature similarity between $j$-th token in frame $f$ and $k$-th token in frame $(f+1)$. Each token links to its most similar counterpart in the previous frame if their similarity exceeds a merging threshold $T_\tau$. This gradually forms redundancy trees that capture fine-grained temporal variations while avoiding merging dissimilar tokens.

**Compress spatiotemporal redundancy.** Once the redundancy trees are constructed, tokens within each tree are aggregated:

$$c^{(i)} = \text{Agg}(\mathcal{T}^{(i)}), \tag{5}$$

---

**Algorithm 1** FlashVID Compression

---

**Require:** Video features $E_v \in \mathbb{R}^{F \times N_v \times d}$; similarity function $\text{sim}(\cdot, \cdot)$; merging threshold $T_\tau$
**Ensure:** Compressed token set $\hat{\mathcal{X}}$

1: **Stage 1: Attention and Diversity-based Token Selection (ADTS)**
2: **for** $f = 1$ to $F$ **do**
3:      Compute pairwise distance $D^{(f)}$, [CLS] attention $A_{[\text{CLS}]}^{(f)}$, event relevance $\bar{\mathbf{S}}_e^{(f)}$
4:      $\mathcal{I}^{(f)} \leftarrow \text{MMDP}(D^{(f)}, A_{[\text{CLS}]}^{(f)}, \bar{\mathbf{S}}_e^{(f)})$
5:      $\mathcal{R}^{(f)} \leftarrow E_v^{(f)} \setminus \mathcal{I}^{(f)}$
6: **end for**
7: **Stage 2: Tree-based Spatiotemporal Token Merging (TSTM)**
8: Initialize each token in $\mathcal{R}^{(f)}$ as a root node and let $\mathcal{C}$ be an empty token set.
9: **for** $f = 2$ to $F$ **do**             ▷ Build spatiotemporal redundancy trees
10:      **for** each token $r_i^f \in \mathcal{R}^{(f)}$ **do**
11:          $p^* \leftarrow \arg\max_{p \in \mathcal{R}^{(f-1)}} \text{sim}(r_i^f, p)$
12:          **if** $\text{sim}(r_i^f, p^*) \geq T_\tau$ **then**
13:              Connect $r_i^f$ to $p^*$
14:          **end if**
15:      **end for**
16: **end for**
17: **for** each tree $\mathcal{T}$ **do**                    ▷ Aggregate redundancy trees
18:      $\mathcal{C} \leftarrow \mathcal{C} \cup \text{Agg}(\mathcal{T})$
19: **end for**
20: **return** $\hat{\mathcal{X}} \leftarrow \mathcal{C} \cup (\bigcup_{f=1}^{F} \mathcal{I}^{(f)})$

---

where $\mathcal{T}^{(i)}$ denotes the $i$-th spatiotemporal redundancy tree and $\text{Agg}(\cdot)$ represents an aggregation function (e.g., mean pooling), producing compact yet informative spatiotemporal representations.

The quality of redundancy trees is critical for fine-grained compression. Although we've explored constraining tree depth and breadth to prevent merging spatiotemporally distant tokens, it yielded negligible gains; thus, no such constraints are applied in practice (see Appendix A.3 for details.)

### 3.3 ATTENTION AND DIVERSITY-BASED TOKEN SELECTION

Although TSTM effectively compresses spatiotemporal redundancy, it may discard important tokens in noisy and high-volume inputs. To mitigate this, we introduce the Attention and Diversity-based Token Selection (ADTS) module, which prioritizes spatiotemporally informative tokens within each frame while ensuring feature diversity for robust video representations. ADTS formulates token selection as a frame-wise Max-Min Diversity Problem (MMDP) (Alvar et al., 2025). Given video features $E_v \in \mathbb{R}^{F \times N_v \times d}$, we first compute the frame-wise cosine distance matrix:

$$D^{(f)} = 1 - \cos(E_v^{(f)}, E_v^{(f)}), \tag{6}$$

where $D^{(f)} \in \mathbb{R}^{N_v \times N_v}$ denotes the pairwise feature dissimilarities in frame $f$. Solving MMDP on $D^{(f)}$ yields a diverse token subset in frame $f$ with the maximal minimum distance. However, diversity alone may overlook the most informative visual tokens. To address this issue, we introduce two calibration terms: 1) **[CLS] attention** and 2) **event relevance**.

**[CLS] attention calibration.** We extract the attention matrices from the vision encoder. For those encoders without an explicit [CLS] token (e.g., SigLIP (Zhai et al., 2023)), we derive it from the attention matrix:

$$A = \text{Softmax}(QK^T/\sqrt{d}) \in \mathbb{R}^{F \times N_v \times N_v}, \tag{7}$$

and compute $A_{[\text{CLS}]} \in \mathbb{R}^{F \times N_v}$ by averaging attention weights each token receives within its frame. This calibration highlights informative tokens in each frame.

**Event relevance calibration.** Event relevance measures a token's correlation with the current video context. We obtain frame embeddings $f_v = \text{GAP}(E_v) \in \mathbb{R}^{F \times d}$ by global average pooling and compute the event similarity matrices:

$$\bar{\mathbf{S}}_e = \frac{1}{F} \sum_{i=1}^{F} (E_v \cdot f_v^\top)[:, :, i] \in \mathbb{R}^{F \times N_v}. \tag{8}$$

This calibration emphasizes the tokens most relevant to the video event. Finally, spatiotemporally informative tokens are selected by solving:

$$\mathcal{I} = \text{MMDP}(D, A_{\textbf{[CLS]}}, \bar{\mathbf{S}}_e). \tag{9}$$

As summarized in Alg. 1, FlashVID compresses video redundancy in two stages: ADTS first selects spatiotemporally informative tokens by solving a calibrated Max-Min Diversity Problem (see Appendix C.2 for details), then TSTM merges redundant tokens across frames through spatiotemporal redundancy trees, yielding compact yet informative visual features.

## 4 EXPERIMENTS

In this section, we conduct extensive experiments across multiple benchmarks and VLLMs. We provide a brief introduction to the experimental settings in Sec. 4.1, present the main experimental results in Sec. 4.2, and discuss essential ablation studies in Sec. 4.3.

### 4.1 EXPERIMENTAL SETTINGS

**Benchmarks.** We evaluate our method on **five** widely-used video understanding benchmarks: VideoMME (Fu et al., 2025a), EgoSchema (Mangalam et al., 2023), LongVideoBench (Wu et al., 2024), MVBench (Li et al., 2024a), and MLVU (Zhou et al., 2025). Notably, these benchmarks cover a wide range of video durations and complex scenarios, providing a comprehensive evaluation of our method's effectiveness and generalization. Additional details can be found in Appendix B.

**Compared baselines.** We compare FlashVID with four state-of-the-art training-free VLLM acceleration methods: 1) **FastV** (Chen et al., 2024), which selects prompt-relevant tokens via text-to-visual attention at the prefilling stage; 2) **VisionZip** (Yang et al., 2025c), pruning tokens using [CLS] attention and spatial merging before the LLM; 3) **PruneVID** (Huang et al., 2025), combining spatiotemporal token merging with attention-based selection in the LLM; and 4) **FastVID** (Shen et al., 2025), compressing redundant tokens via density-based spatiotemporal pruning.

**Implementation details.** We evaluate our method on three representative VLLMs: LLaVA-OneVision (Li et al., 2025a), LLaVA-Video (Zhang et al., 2024), and Qwen2.5-VL (Bai et al., 2025b), which cover diverse architectures to ensure generality. Following the official setting, LLaVA-OneVision and LLaVA-Video uniformly sample 32 and 64 frames, producing $32 \times 196$ and $64 \times 169$ visual tokens, respectively. For LLaVA-Video, we adopt frame token setting, facilitating adaptation for different acceleration frameworks. To ensure a fair comparison, we align the average token budget per transformer layer. Since TSTM compresses redundancy via thresholding, we further apply frame-wise token compression based on DPC-kNN to meet the predefined token budget. Unless otherwise specified, we utilize the **same** set of hyperparameters for all experiments. All the experiments are conducted on NVIDIA A800 80G GPUs using LMMs-Eval (Zhang et al., 2025b). Additional implementation details are provided in Appendix C.

### 4.2 MAIN RESULTS

We evaluate FlashVID against state-of-the-art baselines on three representative VLLMs with distinct architectures under various retention ratios $R$. Additional experimental results on Qwen2.5-VL and LLaVA-Video are reported in Appendix. A.

**Results on LLaVA-OneVision.** Tab. 1 compares FlashVID with other methods on LLaVA-OneVision. VisionZip performs competitively at higher retention (i.e., $25\%, 20\%$) but suffers sharp

Table 1: **Comparison of state-of-the-art methods on LLaVA-OneVision and LLaVA-Video.** Our FlashVID consistently outperforms previous state-of-the-art methods by a large margin under different retention ratios across multiple benchmarks and VLLMs. Notably, FlashVID surpasses vanilla LLaVA-OneVision with full visual tokens input when $R \in \{15\%, 20\%, 25\%\}$.

| Method | Retention Ratio $R$ | VideoMME | | | | EgoSchema | | LongVideo Bench | MVBench | Avg. | |
|---|---|---|---|---|---|---|---|---|---|---|---|
| | | Short | Medium | Long | Overall | Subset | Total | | | Score | Rel. Acc (%) |
| *LLaVA-OneVision* | | | | | | | | | | | |
| Vanilla | 100% | 69.9 | 56.7 | 48.9 | 58.5 | 62.2 | 60.3 | 56.6 | 58.3 | 58.4 | 100.0 |
| FastV | | 68.1 | 54.7 | 46.8 | 56.5 | 60.4 | 57.8 | 55.4 | 56.4 | 56.5 | 96.7 |
| VisionZip | | 68.8 | 57.3 | 48.2 | 58.1 | 63.0 | 60.4 | 56.4 | 57.8 | 58.2 | 99.7 |
| PruneVID | 25% | 67.3 | 54.8 | 47.2 | 56.4 | 61.0 | 58.1 | 55.4 | 56.8 | 56.7 | 97.1 |
| FastVID | | 69.9 | 56.3 | 47.4 | 57.9 | 61.2 | 59.5 | 55.9 | 58.1 | 57.8 | 99.0 |
| FlashVID | | **71.2** | **57.0** | **49.3** | **59.2** | **63.4** | **60.4** | **56.8** | **58.0** | **58.6** | **100.3** |
| FastV | | 66.3 | 53.9 | 46.9 | 55.7 | 60.6 | 57.6 | 56.0 | 56.0 | 56.3 | 96.4 |
| VisionZip | | 68.6 | **57.0** | 48.3 | 58.0 | 62.0 | 60.0 | 55.4 | 57.6 | 57.7 | 98.8 |
| PruneVID | 20% | 67.2 | 53.9 | 48.2 | 56.4 | **63.2** | **60.2** | 55.2 | 56.2 | 57.0 | 97.6 |
| FastVID | | 69.9 | 56.3 | 47.4 | 57.9 | 61.2 | 59.5 | 55.9 | 58.1 | 57.9 | 99.1 |
| FlashVID | | **70.1** | 55.4 | **48.9** | **58.2** | 63.0 | 60.1 | **58.5** | **58.2** | **58.7** | **100.5** |
| FastV | | 64.6 | 54.0 | 45.3 | 54.6 | 59.8 | 56.6 | 54.8 | 55.0 | 55.2 | 94.5 |
| VisionZip | | 63.8 | 54.6 | 48.3 | 55.6 | 62.8 | 60.0 | 54.1 | 53.5 | 55.8 | 95.5 |
| FastVID | 15% | **69.7** | 55.8 | 47.7 | 57.7 | 58.8 | 58.9 | 56.7 | **58.2** | 57.9 | 99.1 |
| PruneVID | | 67.2 | 52.8 | 46.7 | 56.1 | 61.6 | 57.7 | 54.5 | 55.1 | 55.7 | 95.4 |
| FlashVID | | 69.6 | **56.0** | **48.9** | **58.2** | **62.8** | **60.4** | **57.5** | 57.9 | **58.5** | **100.2** |
| FastV | | 60.9 | 52.2 | 44.9 | 52.7 | 59.0 | 56.0 | 52.4 | 53.4 | 53.6 | 91.8 |
| VisionZip | | 60.3 | 52.9 | 46.7 | 53.3 | 61.6 | 58.5 | 49.4 | 54.8 | 54.0 | 92.5 |
| PruneVID | 10% | 65.9 | 52.8 | 45.6 | 54.7 | 60.0 | 57.2 | 54.0 | 53.7 | 54.9 | 94.0 |
| FastVID | | **68.1** | 55.7 | 47.8 | 57.2 | 58.8 | 58.7 | 55.7 | 57.0 | 57.1 | 97.8 |
| FlashVID | | 67.3 | **57.1** | **49.0** | **57.8** | **62.4** | **60.0** | **56.5** | **57.4** | **57.9** | **99.1** |
| *LLaVA-Video* | | | | | | | | | | | |
| Vanilla | 100% | 77.0 | 62.1 | 53.3 | 64.2 | 59.4 | 57.3 | 59.5 | 61.9 | 60.7 | 100.0 |
| FastV | | 69.3 | 58.3 | 49.9 | 59.2 | 54.8 | 54.1 | 56.0 | 58.4 | 56.9 | 93.7 |
| VisionZip | 20% | 72.3 | 59.6 | 53.3 | 61.7 | **59.0** | 56.4 | 58.0 | 59.8 | 59.0 | 97.2 |
| FastVID | | **74.6** | **60.8** | 52.3 | **62.6** | 57.0 | 55.0 | 57.1 | **60.2** | 58.7 | 96.7 |
| FlashVID | | 74.1 | 60.0 | **52.3** | 62.2 | 58.4 | **56.4** | **58.7** | 59.8 | **59.3** | **97.7** |
| FastV | | 64.3 | 53.8 | 49.2 | 55.8 | 50.6 | 51.1 | 53.6 | 56.2 | 54.2 | 89.3 |
| VisionZip | 10% | 69.4 | 57.9 | 51.2 | 59.5 | 54.4 | 53.9 | 54.5 | 58.5 | 56.6 | 93.2 |
| FastVID | | 71.8 | 57.3 | 50.2 | 59.8 | 54.8 | 52.4 | 56.9 | 59.3 | 57.1 | 94.1 |
| FlashVID | | **72.2** | **59.1** | **51.2** | **60.9** | **57.2** | **54.9** | **57.7** | **59.3** | **58.2** | **95.9** |

degradation at $15\%, 10\%$ due to excessive loss from aggressive spatial compression. FastV shows the weakest performance, as early-layer pruning is unstable. In contrast, FlashVID achieves the best results across all retention ratios, preserving **99.1%** of the vanilla model's accuracy even at $R = 10\%$. Moreover, when $R \in \{25\%, 20\%, 15\%\}$, FlashVID surpasses the vanilla LLaVA-OneVision with full visual tokens input, revealing a *"less is more"* pattern where excessively redundant tokens may degrade performance.

**Results on LLaVA-Video.** LLaVA-Video employs a specialized design by inserting newline tokens to inject spatiotemporal positional information. Unlike the official grid token setting, we apply the frame token in LLaVA-Video, which facilitates adaptation for different acceleration frameworks, where we found that these two settings lead to similar performance. In Tab. 1, we evaluate our method against other methods on LLaVA-Video. Notably, our FlashVID outperforms all baselines under various retention ratios.

**Results on Qwen2.5-VL** In addition to LLaVA-OneVision and LLaVA-Video, we also evaluate our FlashVID against other methods on Qwen2.5-VL, which shows significantly different archi-

Table 2: **Comparison of state-of-the-art methods on Qwen2.5-VL.** The best performance among those with similar retention ratios $R$ is highlighted in bold.

| Method | Retention Ratio $R$ | VideoMME | | | | EgoSchema | | LongVideo Bench | MVBench | Avg. | |
|---|---|---|---|---|---|---|---|---|---|---|---|
| | | Short | Medium | Long | Overall | Subset | Total | | | Score | Rel. Acc (%) |
| Vanilla | 100% | 72.6 | 61.4 | 49.9 | 61.3 | 60.2 | 58.3 | 58.9 | 68.0 | 61.6 | 100.0 |
| FastV | 20% | 69.4 | 57.0 | **51.2** | 59.2 | **60.2** | **57.1** | 54.2 | **66.8** | 59.3 | 96.3 |
| VisionZip | | 69.6 | 57.2 | 50.2 | 59.0 | 58.6 | 56.6 | 56.3 | 66.4 | 59.6 | 96.8 |
| FastVID | | 69.9 | 56.3 | 49.7 | 58.6 | 57.2 | 56.4 | 57.8 | 64.7 | 59.4 | 96.4 |
| FlashVID | | **70.4** | **58.6** | 49.7 | **59.6** | 59.2 | 56.8 | **58.1** | 66.5 | **60.2** | **97.7** |
| FastV | 10% | 63.7 | 54.7 | 49.3 | 55.9 | 58.6 | 54.9 | 51.1 | 63.6 | 57.3 | 91.6 |
| VisionZip | | 67.0 | 54.7 | 47.6 | 56.4 | 55.8 | 55.5 | 54.5 | 64.3 | 57.7 | 93.7 |
| FastVID | | 66.3 | 53.6 | 49.0 | 56.3 | 56.0 | 55.6 | 55.4 | 62.3 | 57.4 | 93.2 |
| FlashVID | | **68.1** | **54.7** | **49.0** | **57.3** | **57.4** | **55.9** | **57.1** | **65.5** | **58.9** | **95.6** |

Table 3: **Comparison of state-of-the-art methods on Qwen2.5-VL under a fixed token budget.** Our FlashVID enables Qwen2.5-VL processing **10×** video frames, improving the overall performance of **8.6**% within the same computational memory budget.

| Method | #Frames | Retention Ratio $R$ | VideoMME | | | | EgoSchema | | LongVideo Bench | MLVU | Avg. | |
|---|---|---|---|---|---|---|---|---|---|---|---|---|
| | | | Short | Medium | Long | Overall | Subset | Total | | | Score | Rel. Acc (%) |
| Vanilla | 16 (1x) | 100% | 66.4 | 56.4 | 48.2 | 57.0 | 58.2 | 55.6 | 56.9 | 40.6 | 52.6 | 100.0 |
| VisionZip | 80 (5x) | 20% | 74.2 | 60.0 | 52.1 | 62.1 | 60.0 | 58.2 | 57.4 | 43.1 | 55.2 | 104.9 |
| FastVID | | | 73.0 | 59.9 | 51.7 | 61.5 | 61.2 | 58.4 | 58.0 | 44.4 | 55.6 | 105.7 |
| FlashVID | | | **74.2** | **60.8** | **52.2** | **62.4** | **61.4** | **58.6** | **58.9** | **45.0** | **56.2** | **106.8** |
| VisionZip | 160 (10x) | 10% | 70.7 | 60.1 | **53.9** | 61.6 | **61.8** | **59.6** | 56.8 | 45.1 | 55.8 | 106.1 |
| FastVID | | | 71.2 | 60.6 | 53.8 | 61.9 | 61.2 | 59.1 | 58.0 | 43.8 | 55.7 | 105.9 |
| FlashVID | | | **71.4** | **62.2** | 53.7 | **62.4** | 61.2 | 59.5 | **58.9** | **47.5** | **57.1** | **108.6** |

tecture and characteristics. As illustrated in Tab. 2, our method significantly surpasses previous state-of-the-art methods under various retention ratios, demonstrating strong generalization across different VLLMs.

**Results on Qwen2.5-VL under fixed token budget.** Due to computational and memory constraints, existing VLLMs typically process only a small number of sampled frames, often missing important visual cues. To assess the benefit of longer temporal context under a fixed computational budget, we apply token compression to enable models to process more frames. As shown in Tab. 3, Qwen2.5-VL achieves consistent improvements over its vanilla 16-frame baseline when equipped with token compression frameworks. Among them, FlashVID delivers the largest performance gains, highlighting its ability to unlock longer video sequences and demonstrating superior efficiency in constrained settings.

## 4.3 ABLATION STUDIES

In this section, we conduct ablation studies on the ADTS module and the retained ratio $\alpha$ of ADTS and TSTM using LLaVA-OneVision. Additional ablation studies are provided in Appendix. A.3.

**Ablation study on ADTS module.** ADTS is proposed to select both important and diverse tokens. As shown in Tab. 4, we compare our ADTS with ATS and DTS, i.e., attention-based and diversity-based token selection. Our ADTS outperforms other token selection methods based solely on [CLS] attention (ATS) and feature diversity (DTS) by a large margin, demonstrating that ADTS can effectively identify the important visual tokens.

To realize a comprehensive ablation study on ADTS, we further ablate the calibration terms used in ADTS. Tab. 4 reveals that both [CLS] attention and event relevance calibration improve performance, while the optimal performance is yielded at the combination of the two.

Table 4: **Ablation study on ADTS.** ATS, DTS, and ADTS denote attention-, diversity-, and attention-diversity-based token selection, respectively. 'C.A' and 'E.R.' denote [CLS] attention and event relevance calibration terms in ADTS.

| Method | VideoMME | EgoSchema | LongVideo Bench | MVBench | Rel. Acc. |
|--------|----------|-----------|-----------------|---------|-----------|
| ATS | 55.5 | 59.5 | 55.0 | 56.2 | 96.9 |
| DTS | 55.7 | **60.3** | 55.3 | 55.5 | 97.1 |
| w/ E.R. | 56.0 | 60.2 | 55.1 | 56.8 | 97.6 |
| w/ C.A. | 57.3 | 59.7 | 55.7 | 57.3 | 98.5 |
| ADTS | **57.8** | 60.0 | **56.5** | **57.4** | **99.1** |

Table 5: **Ablation study on $\alpha$ in visual token compression before LLM.** $\alpha$ controls the retained ratio of ADTS and TSTM, where $\alpha = 0$ and $\alpha = 1$ indicate TSTM and ADTS only.

| $\alpha$ | VideoMME | EgoSchema | LongVideo Bench | MVBench | Rel. Acc. |
|----------|----------|-----------|-----------------|---------|-----------|
| 0.0/TSTM | 56.7 | 60.2 | 55.3 | 55.6 | 97.4 |
| 0.2 | 56.2 | 59.8 | 55.3 | 56.5 | 97.4 |
| 0.4 | 56.4 | 60.0 | 55.1 | 57.2 | 97.9 |
| 0.6 | 57.0 | **60.4** | 55.8 | 57.0 | 98.5 |
| 0.7 | **57.8** | 60.0 | **56.5** | 57.4 | **99.1** |
| 0.8 | 57.2 | 60.1 | 56.3 | 57.1 | 98.8 |
| 1.0/ADTS | 56.9 | 60.1 | 55.6 | **57.6** | 98.5 |

Table 6: **Efficiency of our FlashVID.** We conduct the efficiency analysis on LLaVA-OneVision, and report the prefilling time and Time-To-First-Token (TTFT) in milliseconds (ms).

| Method | Retention Ratio $R$ | TFLOPs | Vision Encoding | Prefilling Time | | | TTFT | Avg. | |
|--------|---------------------|--------|-----------------|-----------------|--|--|------|------|--|
| | | | | Compression | LLM Forward | Total | | Score | Rel. Acc (%) |
| Vanilla | 100% | 113.4 | 785.0 | - | 1220.8 | 1220.8 (1.0×) | 2005.8 (1.0×) | 58.9 | 100.0 |
| FastVID | 25% | 22.4 | 785.0 | 28.6 | 273.2 | 301.8 (4.0×) | 1086.8 (1.8×) | 58.0 | 98.5 |
| FlashVID | 10% | 8.6 | 785.0 | 60.2 | 133.1 | 193.3 (**6.3**×) | 978.3 (**2.1**×) | 58.4 | **99.1** |

**Ablation study on $\alpha$.** As illustrated in Tab. 5, we conduct an ablation study on merging threshold $\alpha$, which controls the ratio of visual tokens retained between ADTS and TSTM compression. In particular, $\alpha = 0$ and $\alpha = 1$ denote TSTM and ADTS only, respectively. The experimental results show that ADTS alone ($\alpha = 1$) outperforms TSTM alone ($\alpha = 0$). However, the peak performance is achieved at $\alpha = 0.7$, implying that a balanced integration of these two modules (i.e., ADTS and TSTM) is necessary to maintain the model performance.

## 4.4 EFFICIENCY ANALYSIS

Although token compression can effectively improve the inference efficiency of VLLMs, it can also be a time-consuming operation. In Tab. 6, we conduct an efficiency experiment on LLaVA-OneVision using a single NVIDIA A100 GPU compared to FastVID on VideoMME. Remarkably, FlashVID preserves **99.1**% relative accuracy at $R = 10\%$, while FastVID achieves a similar performance at $R = 25\%$. Consequently, FlashVID enables **6.3**× prefilling and **2.1**× Time-To-First-Token (TTFT) speedups, largely outperforming FastVID. Additional efficiency experimental results on LLaVA-Video can be found in Appendix. A.2.

## 5 CONCLUSION

In this work, we introduce FlashVID, a training-free and plug-and-play acceleration framework for VLLMs. FlashVID combines Attention and Diversity-based Token Selection (ADTS) for representative token filtering with Tree-based Spatiotemporal Token Merging (TSTM) for fine-grained redundancy elimination, effectively compressing spatiotemporal redundancy while preserving essential visual information. Extensive experiments on three VLLMs across five video understanding benchmarks demonstrate that FlashVID achieves superior performance in both efficiency and accuracy. In particular, it can serve as a plug-and-play module, enabling VLLMs to process significantly longer video sequences under a constrained computational budget.

## ACKNOWLEDGEMENT

This work was supported by the Guangdong Basic and Applied Basic Research Foundation (2025A1515011546) and by the Shenzhen Science and Technology Program (JCYJ20240813105901003, KJZD20240903102901003, ZDCY20250901113000001).

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

# Supplementary Material

CONTENTS

Table 7: **Comparison of state-of-the-art methods on Qwen2.5-VL.** The best performance among those with similar retention ratios $R$ is highlighted in bold.

| Method | Retention Ratio $R$ | VideoMME | | | | EgoSchema | | LongVideo Bench | MVBench | Avg. | |
|---|---|---|---|---|---|---|---|---|---|---|---|
| | | Short | Medium | Long | Overall | Subset | Total | | | Score | Rel. Acc (%) |
| Vanilla | 100% | 72.6 | 61.4 | 49.9 | 61.3 | 60.2 | 58.3 | 58.9 | 68.0 | 61.6 | 100.0 |
| FastV | 25% | 71.2 | 57.8 | **51.6** | **60.2** | **60.2** | 57.2 | 54.6 | **67.4** | 59.8 | 97.1 |
| VisionZip | | 70.7 | 57.9 | 51.2 | 59.9 | 58.6 | 57.0 | 57.1 | 67.3 | 60.3 | 97.9 |
| FastVID | | **71.2** | 57.8 | 49.9 | 59.6 | 58.2 | 56.7 | 58.0 | 65.5 | 60.0 | 97.4 |
| FlashVID | | 71.1 | **58.7** | 49.1 | 59.6 | 59.4 | **57.2** | **58.1** | 67.1 | **60.5** | **98.2** |
| FastV | 15% | 67.4 | 55.2 | **51.1** | 57.9 | **59.6** | 56.5 | 52.2 | 65.9 | 58.1 | 94.3 |
| VisionZip | | 68.8 | 56.7 | 49.1 | 58.2 | 56.2 | 55.7 | 56.3 | 66.0 | 59.0 | 95.8 |
| FastVID | | 68.2 | 56.9 | 49.4 | 58.2 | 56.6 | 56.0 | 56.8 | 63.6 | 58.6 | 95.1 |
| FlashVID | | **69.8** | **57.1** | 49.3 | **58.7** | 57.6 | 56.4 | **56.8** | **66.6** | **59.6** | **96.8** |

Table 8: **Comparison of state-of-the-art methods on Qwen2.5-VL under a fixed token budget.** Our FlashVID enables Qwen2.5-VL processing **10×** video frames, improving the overall performance of **8.6**% within the same computational memory budget.

| Method | #Frames | Retention Ratio $R$ | VideoMME | | | | EgoSchema | | LongVideo Bench | MLVU | Avg. | |
|---|---|---|---|---|---|---|---|---|---|---|---|---|
| | | | Short | Medium | Long | Overall | Subset | Total | | | Score | Rel. Acc (%) |
| Vanilla | 16 (1x) | 100% | 66.4 | 56.4 | 48.2 | 57.0 | 58.2 | 55.6 | 56.9 | 40.6 | 52.6 | 100.0 |
| VisionZip | 48 (3x) | 33.3 | **74.0** | 59.6 | **52.9** | **62.2** | 59.4 | 57.6 | 56.1 | 42.2 | 54.5 | 103.6 |
| FastVID | | | 73.2 | 60.0 | 51.7 | 61.6 | 59.4 | 57.8 | 57.2 | 43.1 | 54.9 | 104.4 |
| FlashVID | | | 73.0 | **60.2** | 52.4 | 61.9 | **59.6** | **57.8** | **57.0** | **45.1** | **55.4** | **105.3** |
| VisionZip | 64 (4x) | 25.0 | 72.3 | **60.1** | **51.6** | **61.3** | 61.0 | **58.5** | 57.8 | 44.7 | 55.6 | 105.7 |
| FastVID | | | 71.0 | 58.3 | 50.6 | 60.0 | **61.4** | 58.1 | 57.7 | 45.0 | 55.2 | 104.9 |
| FlashVID | | | **73.0** | 59.3 | 50.3 | 60.9 | 60.2 | 58.4 | **58.4** | 45.0 | **55.7** | **105.9** |

# A    MORE EXPERIMENTAL RESULTS

We present comprehensive experimental results of our method. In the Appendix. A.1, we evaluate our FlashVID against previous state-of-the-art-methods on Qwen2.5-VL (Bai et al., 2025b). In the Appendix. A.2, we present additional experimental results on LLaVA-Video. In the Appendix. A.3, we provide additional ablation studies on FlashVID.

## A.1    ADDITIONAL EXPERIMENTS ON QWEN2.5-VL

**Results on Qwen2.5-VL.**    To further demonstrate the generalizability of our method, we evaluate it against other methods on Qwen2.5-VL (Bai et al., 2025b), which shows significant differences relative to LLaVA-OneVision and LLaVA-Video. Tab. 2 presents a part of the experimental results on Qwen2.5-VL under retention ratios $R \in \{20\%, 10\%\}$. Additional experimental results when $R \in \{25\%, 15\%\}$ on Qwen2.5-VL are provided in Tab. 7. Notably, our method consistently surpasses previous state-of-the-art methods under various retention ratios, demonstrating strong generalization across different VLLMs.

**Results on Qwen2.5-VL under fixed token budget.**    By applying visual token compression, VLLMs can achieve performance gains by processing more video frames while maintaining the overall computational budget. As discussed in Sec. 4.2, we explore extending the number of input frames under a fixed token budget. Tab. 3 reports results with $5\times$ and $10\times$ frames, demonstrating that VLLMs benefit from longer temporal context without increasing computational cost. Additional results with $3\times$ and $4\times$ frames are presented in Tab. 8, revealing a consistent improvement trend. It highlights that FlashVID effectively compresses visual tokens and preserves compact yet informative representations.

Table 9: **Comparison of state-of-the-art methods on LLaVA-Video.** We employ frame token setting for adaptation to different acceleration frameworks.

| Method | Retention Ratio $R$ | VideoMME | | | | EgoSchema | | LongVideo Bench | MVBench | Avg. | |
|---|---|---|---|---|---|---|---|---|---|---|---|
| | | Short | Medium | Long | Overall | Subset | Total | | | Score | Rel. Acc (%) |
| Vanilla | 100% | 77.0 | 62.1 | 53.3 | 64.2 | 59.4 | 57.3 | 59.5 | 61.9 | 60.7 | 100.0 |
| FastV | 25% | 71.7 | 59.2 | 50.9 | 60.6 | 56.0 | 54.8 | 56.4 | 59.1 | 57.7 | 95.1 |
| VisionZip | | 74.0 | 60.3 | 52.9 | 62.4 | 59.0 | **57.0** | 58.3 | 60.0 | 59.4 | 97.9 |
| FastVID | | **74.7** | 60.1 | **53.6** | **62.8** | 57.4 | 55.4 | 58.2 | **60.5** | 59.2 | 97.5 |
| FlashVID | | 74.2 | **61.4** | 51.6 | 62.4 | **59.2** | 56.6 | **59.1** | 60.2 | **59.6** | **98.2** |
| FastV | 15% | 67.9 | 56.9 | 50.8 | 58.5 | 52.8 | 53.1 | 54.5 | 57.5 | 55.9 | 92.1 |
| VisionZip | | 72.9 | 58.1 | 51.9 | 61.0 | **58.6** | 55.7 | 57.2 | 59.6 | 58.4 | 96.2 |
| FastVID | | 73.4 | 58.1 | 51.8 | 61.1 | 56.8 | 54.1 | 57.7 | 60.3 | 58.3 | 96.0 |
| FlashVID | | **73.8** | **59.6** | **52.1** | **61.8** | 57.8 | **55.8** | **58.3** | **60.4** | **59.1** | **97.4** |

Table 10: **Efficiency of our FlashVID.** We conduct the efficiency analysis on LLaVA-Video, and report the prefilling time and Time-To-First-Token (TTFT) in milliseconds (ms).

| Method | Retention Ratio $R$ | TFLOPs | Vision Encoding | Prefilling Time | | | TTFT | Avg. | |
|---|---|---|---|---|---|---|---|---|---|
| | | | | Compression | LLM Forward | Total | | Score | Rel. Acc (%) |
| Vanilla | 100% | 94.8 | 685.0 | - | 1016.8 | 1016.8 (1.0×) | 1701.8 (1.0×) | 60.7 | 100.0 |
| FlashVID | 10% | 8.0 | 685.0 | 85.7 | 107.6 | 193.3 (**5.3×**) | 878.3 (**1.9×**) | 58.2 | 95.9 |

## A.2 ADDITIONAL EXPERIMENTS ON LLAVA-VIDEO

**Results on LLaVA-Video.** In Tab. 1, we present a part of the experimental results on LLaVA-Video (Zhang et al., 2024) under retention ratios $R \in \{20\%, 10\%\}$. Additional experimental results when $R \in \{25\%, 15\%\}$ on LLaVA-Video are provided in Tab. 9. Notably, FlashVID consistently outperforms previous state-of-the-art methods by a large margin under different retention ratios.

**Additional efficiency analysis** As illustrated in Tab. 6, we test the efficiency of our FlashVID on LLaVA-OneVision (Li et al., 2025a), comparing to FastVID (Shen et al., 2025). In Tab. 10, we further evaluate the efficiency of our FlashVID on LLaVA-Video (Zhang et al., 2024). We report the detailed prefilling time and Time-To-First-Token (TTFT). Notably, our FlashVID enables **5.3×** prefilling and **1.9×** Time-to-First-Token (TTFT) speedups over the vanilla LLaVA-Video while maintaining **95.9%** relative accuracy at 10% retention ratio.

## A.3 ADDITIONAL ABLATION STUDIES

**Ablation study on $T_\tau$.** The merging threshold $T_\tau$ plays an important role in the Tree-based Spatiotemporal Token Merging (TSTM) module. $T_\tau$ directly influences the compression quality, in which increasing $T_\tau$ reduces the merging strength and better preserves temporal details, whereas lowering $T_\tau$ promotes aggressive compression but may merge less correlated tokens, probably introducing noise to the compact representation. As illustrated in Tab. 13, we conduct an ablation study on merging threshold $T_\tau$ on LLaVA-OneVision and LLaVA-Video at $R = 10\%$. FlashVID consistently achieves the best performance under different VLLMs when merging threshold $T_\tau = 0.8$.

**Ablation study on tree depth and breadth constraints.** In TSTM, video redundancy is jointly modeled by spatiotemporal redundancy trees, which connect the highly correlated spatiotemporal visual information. Intuitively, applying proper depth and breadth constraints avoids the merge of spatiotemporally distant tokens, which may improve the compression quality. In addition to the threshold parameter $T_\tau$, we also test with two extra parameters: 1) *depth constraint*: aims to maintain the temporal dynamics, preventing tokens from spanning an excessively long temporal range in the same tree; 2) *breadth constraint*: seeks to preserve the spatial locality, avoiding merging across overly large spatial regions. The detailed implementation of TSTM with depth and breadth constraints is provided in Alg. 2.

Table 11: **Ablation study on the tree depth.** The maximum tree depth constraint prevents token merging in tokens from spanning an overly long temporal range.

| Depth | VideoMME | EgoSchema | LongVideo Bench | MVBench | Rel. Acc. |
|---|---|---|---|---|---|
| 1/Min | 57.2 | 60.0 | 55.5 | 57.2 | 98.5 |
| 4 | 57.6 | **60.3** | 56.4 | 57.2 | 99.1 |
| 8 | 57.6 | 60.0 | 56.0 | 57.4 | 99.0 |
| 16 | 57.7 | 60.0 | 56.3 | 57.3 | 99.0 |
| 32/Inf | **57.8** | 60.0 | **56.5** | 57.4 | **99.1** |

Table 12: **Ablation study on tree breadth.** The maximum tree breadth prevents the merge of tokens in adjacent frames from crossing excessively large spatial regions, ensuring that spatial locality is preserved.

| Breadth | VideoMME | EgoSchema | LongVideo Bench | MVBench | Rel. Acc. |
|---|---|---|---|---|---|
| 1/Min | 57.3 | 60.1 | 56.0 | 57.2 | 98.6 |
| 5 | 57.3 | **60.1** | 56.0 | 57.2 | 98.8 |
| 9 | **57.9** | 60.0 | 56.0 | 56.8 | 98.8 |
| 14/Inf | 57.8 | 60.0 | **56.5** | 57.4 | **99.1** |

Table 13: **Ablation study on the $T_\tau$.** $T_\tau$ controls the merging strength, in which a lower $T_\tau$ indicates stronger compression.

| $T_\tau$ | VideoMME | EgoSchema | LongVideo Bench | MVBench | Rel. Acc. |
|---|---|---|---|---|---|
| | | *LLaVA-OneVision* | | | |
| 0.9 | 57.3 | **60.4** | 56.5 | 57.3 | 99.1 |
| 0.8 | **57.8** | 60.0 | **56.5** | **57.4** | **99.1** |
| 0.7 | 57.1 | 60.1 | 56.0 | 57.0 | 98.5 |
| | | *LLaVA-Video* | | | |
| 0.9 | 57.2 | **55.0** | 56.9 | **59.7** | 95.7 |
| 0.8 | 57.2 | 54.9 | **57.7** | 59.3 | **95.9** |
| 0.7 | **57.8** | 55.3 | 57.1 | 59.2 | 95.7 |

Table 14: **Ablation study on $f_e$.** $f_e$ controls the expansion ratio, in which a large $f_e$ may lead to computational inefficiency, while a low value may lose critical information.

| $f_e$ | VideoMME | EgoSchema | LongVideo Bench | MVBench | Rel. Acc. |
|---|---|---|---|---|---|
| 1.00 | 56.5 | 60.4 | 55.1 | 56.5 | 97.8 |
| 1.15 | 56.9 | 60.2 | 55.4 | 56.9 | 98.1 |
| 1.20 | 57.3 | 60.3 | 56.0 | 57.3 | 98.8 |
| 1.25 | **57.8** | 60.0 | **56.5** | 57.4 | **99.1** |
| 1.30 | 57.5 | **60.3** | 56.3 | **57.5** | 99.1 |
| 1.35 | 57.1 | 60.0 | 56.5 | 57.3 | 98.8 |

However, as illustrated in Tab. 11 and Tab. 12, we conduct ablation studies on tree depth and breadth using LLaVA-OneVision. Experimental results show that depth and breadth constraints don't bring performance gains. We hypothesize that the merging threshold $T_\tau$ delivers a similar effect.

**Ablation study on $f_e$.** FlashVID retains more visual tokens input to the LLM while pruning within the LLM to satisfy the overall computational budget, avoiding the loss of important visual information. $f_e$ controls the expansion ratio, in which a large $f_e$ may lead to computational inefficiency, while a low value may lose critical information. In Tab. 14, we conduct an ablation study on $f_e$ on LLaVA-OneVision. FlashVID achieves the best performance (**99.1%** relative accuracy) when the expansion factor $f_e \in \{1.25, 1.30\}$. We adopt $f_e = 1.25$ for better efficiency.

# B  EVALUATION BENCHMARKS

The experiments are conducted on the following widely used video understanding benchmarks.

**VideoMME.** VideoMME (Fu et al., 2025a) is a comprehensive multi-modal evaluation benchmark on video understanding capabilities of VLLMs. It features 900 videos spanning 6 diverse domains and 30 subcategories, with durations ranging from 11 seconds to 1 hour. Each video is accompanied by high-quality human annotations, including 2,700 multiple-choice question-answer pairs.

**LongVideoBench.** LongVideoBench (Wu et al., 2024) is a comprehensive benchmark designed to evaluate VLLMs on long-context, interleaved video-language understanding. It characterizes 3,763 videos with durations ranging from 8 seconds to 1 hour. This benchmark comprises 6,678 human-annotated multiple-choice questions based on a novel "referring-reasoning" task, where models must retrieve and reason over specific multimodal contexts referenced in the questions, categorized into 17 fine-grained types across perception and relation levels.

**MVBench.** MVBench (Li et al., 2024a) is a comprehensive benchmark designed to evaluate temporal understanding in multi-modal video tasks, addressing the limitations of existing image-focused

---

**Algorithm 2** Tree-based Spatiotemporal Token Merging with Depth and Breadth Constraints

---

**Require:** Token sequences $\{\mathcal{R}^{(1)}, \mathcal{R}^{(2)}, \dots, \mathcal{R}^{(F)}\}$ from $F$ frames; similarity function $\text{sim}(\cdot, \cdot)$; tree depth function $\text{depth}(\cdot)$; merging threshold $T_\tau$; max tree depth $d_{\max}$; neighborhood size $k$
**Ensure:** Compressed token set $\mathcal{C}$

1: Initialize each token in $\mathcal{R}^{(f)}$ as a root node and let $\mathcal{C}$ be an empty token set.
2: **for** $f = 2$ to $F$ **do**                             ▷ Construct candidate edges
3:      **for** each token $r_i^f \in \mathcal{R}^{(f)}$ **do**
4:          $\mathcal{N}(r_i^f) \leftarrow$ candidate parents in $\mathcal{R}^{(f-1)}$ within neighborhood $k$
5:          $p^* \leftarrow \arg\max_{p \in \mathcal{N}(r_i^f)} \text{sim}(r_i^f, p)$
6:          **if** $\text{sim}(r_i^f, p^*) \geq T_\tau$ **then**
7:              Connect $r_i^f$ to $p^*$
8:          **end if**
9:      **end for**
10: **end for**
11: **for** $f = F$ down to $2$ **do**                          ▷ Backward depth pruning
12:      **for** each token $r_i^f \in \mathcal{R}^{(f)}$ **do**
13:          **if** $\text{depth}(r_i^f) = d_{\max}$ **then**
14:              Disconnect $r_i^f$ from its parent
15:              Mark $r_i^f$ as a new root
16:          **end if**
17:      **end for**
18: **end for**
19: **for** each tree $\mathcal{T}$ **do**                            ▷ Aggregate redundancy trees
20:      $\mathcal{C} \leftarrow \mathcal{C} \cup \text{Agg}(\mathcal{T})$
21: **end for**
22: **return** $\mathcal{C}$

---

benchmarks. It consists of 20 systematically constructed tasks that require complex temporal reasoning skills, generated via static-to-dynamic transformation of static tasks.

**EgoSchema.** EgoSchema (Mangalam et al., 2023) consists of approximately 5,000 five-choice multiple-choice questions derived from 250 hours of egocentric video. It emphasizes long-form temporal reasoning, as each of its 289 three-minute clips requires tracking objects and actions over time spans that are 5–10× longer than those in previous datasets, thereby posing significant challenges for both spatial perception and extended temporal coherence.

**MLVU.** MLVU (Zhou et al., 2025) contains 3,102 multiple-choice questions across nine diverse long-video understanding tasks. It challenges models with videos ranging from 3 minutes to 2 hours, requiring reasoning over plot, temporal order, and event retrieval, thereby jointly testing fine-grained spatial recognition and long-range temporal reasoning.

## C  IMPLEMENTATION DETAILS

### C.1  REPRODUCTION DETAILS OF COMPARED BASELINES

Unless otherwise specified, all the experiments are conducted on NVIDIA A800 80G GPUs on LMMs-Eval (Zhang et al., 2025b) [1]. We evaluate all methods on three representative VLLMs with distinct architectures and characteristics: LLaVA-OneVision (Li et al., 2025a) and LLaVA-Video (Zhang et al., 2024) [2], and Qwen2.5-VL (Bai et al., 2025b) [3]. All baseline methods are reimplemented in LMMs-Eval, following their official implementations:

---

[1] `https://github.com/EvolvingLMMs-Lab/lmms-eval`, MIT License
[2] `https://github.com/LLaVA-VL/LLaVA-NeXT`, Apache License 2.0
[3] `https://github.com/QwenLM/Qwen2.5-VL`, Apache License 2.0

- **FastV** (Chen et al., 2024) [4] **(ECCV 2024)**. FastV prunes tokens at the $K$-th layer of the LLM using cross-modal attention scores, with a pruning ratio $r$. We follow the official settings with $K = 2$, using $r \in \{75\%, 80\%, 85\%, 90\%\}$ for LLaVA-OneVision in Tab. 1 and Qwen2.5-VL in Tab. 2, while setting $r \in \{80\%, 90\%\}$ for LLaVA-Video in Tab. 1.

- **VisionZip** (Yang et al., 2025c) [5] **(CVPR 2025)**. VisionZip prunes visual tokens at the output of the vision encoder, conflicting with pooling operations in VLLMs and resulting in performance degradation. Following (Shen et al., 2025), we instead apply pruning after pooling for VisionZip. We follow the official setting by retaining both dominant and contextual ratios at a 54:10 ratio in each frame. We set $R$ to $\{25\%, 20\%, 15\%, 10\%\}$ for LLaVA-OneVision in Tab. 1 and Qwen2.5-VL in Tab. 2, while setting $R$ to $\{20\%, 10\%\}$ and $\{25\%, 15\%\}$ for LLaVA-Video in Tab. 1 and Tab. 9, respectively.

- **PruneVID** (Huang et al., 2025) [6] **(ACL 2025)**. PruneVID contains both before-LLM compression and inner-LLM pruning during the prefilling stage, along with a KV Cache compression at the decoding stage. Following the official settings, we set the threshold $\tau = 0.8$, the temporal segment ratio $\gamma = 0.25$, the token selection ratio $\alpha = 0.4$, and the pruning layer $K = 10$. We control the token budget by cluster ratio $\beta$. We use $\beta \in \{40.7\%, 32.5\%, 24.4\%, 16.3\%\}$ in Tab. 1.

- **FastVID** (Shen et al., 2025) [7] **(NeurIPS 2025)**. FastVID prunes visual tokens based on spatiotemporal DPC-kNN. It begins with a dynamic segmentation based on transition similarities, followed by a frame-wise salient token selection based on [CLS] attention scores. Finally, it compresses the remaining tokens by spatiotemporal redundancy elimination based on DPC-kNN. Following the official settings, we set the minimum number of segments $c = 8$, the segment threshold $\tau = 0.9$, the salient token ratio $d = 0.4$, the anchor frame step $p = 4$, and the merging factor $\alpha = 0.6$. We set $R$ to $\{25\%, 20\%, 15\%, 10\%\}$ for LLaVA-OneVision in Tab. 1 and Qwen2.5-VL in Tab. 2, while setting $R$ to $\{20\%, 10\%\}$ and $\{25\%, 15\%\}$ for LLaVA-Video in Tab. 1 and Tab. 9, respectively.

### C.2 REPRODUCTION DETAILS OF FLASHVID

In addition to ADTS and TSTM modules, FlashVID employs two design choices: 1) video partition and 2) Inner-LLM Pruning for better performance.

**Video Partition.** State-of-the-art VLLM acceleration methods (Shen et al., 2025; Shao et al., 2025a; Huang et al., 2025; Tao et al., 2025) commonly apply video partitioning before token compression, aiming to avoid information mixing and building upon DySeg (Shen et al., 2025), FlashVID partitions consecutive similar frames into the same segment based on the transition similarities. Instead of using [CLS] token embeddings, we compute transition similarities based on pooled video features. Given video features $E_v \in \mathbb{R}^{F \times N_v \times d}$, we apply global average pooling to obtain the frame embeddings:

$$f_e = \text{GAP}(E_v) \in \mathbb{R}^{F \times d}. \tag{10}$$

The transition similarities are defined as the cosine similarity of frame embeddings of adjacent frames:

$$\begin{aligned} t_i &= \cos(\mathbf{f}_i, \mathbf{f}_{i+1}), \quad i = 1, 2, \cdots, F - 1, \\ \mathbf{T} &= \{t_1, t_2, \cdots, t_{F-1}\} \end{aligned} \tag{11}$$

where $t_i$ denotes the transition similarity between $i$-th and $(i+1)$-th frame. A low transition similarity indicates a significant scene change. Following DySeg, we set the segment threshold $S_\tau = 0.9$ and the minimum number of segments $M_s = 8$.

**Calibrated Max-Min Diversity Problem.** As discussed in Sec. 3.3, FlashVID utilizes the Attention and Diversity-based Token Selection (ADTS) module to identify the spatiotemporally informative tokens within each frame. Specifically, ADTS formulates frame-wise token selection as

---

[4] https://github.com/pkunlp-icler/FastV

[5] https://github.com/JIA-Lab-research/VisionZip, Apache License 2.0

[6] https://github.com/Visual-AI/PruneVid, CC BY-NC-SA 4.0 License

[7] https://github.com/LunarShen/FastVID, MIT License

---

**Algorithm 3** Calibrated Max-Min Diversity Problem (MMDP)

---

**Require:** Pairwise distance $D^{(f)} \in \mathbb{R}^{N_v \times N_v}$; [CLS] attention $A_{[CLS]}^{(f)} \in \mathbb{R}^{N_v}$; event relevance $\bar{\mathbf{S}}_e^{(f)} \in \mathbb{R}^{N_v}$

**Ensure:** Spatiotemporally informative token indices $\mathcal{I}^{(f)}$

1: Initialize selected indices $\mathcal{I}^{(f)} \leftarrow \emptyset$ and $R \leftarrow \{0, 1, ... N_v - 1\}$
2: Let $\mathbf{1}_{N_v} \in \mathbb{R}^{N_v}$ be an all-ones vector
3: $D^{(f)} \leftarrow D^{(f)} \odot \left( \left( A_{[CLS]}^{(f)} \otimes \mathbf{1}_{N_v} \right) \odot \left( \bar{\mathbf{S}}_e^{(f)} \otimes \mathbf{1}_{N_v} \right) \right)$      ▷ Calibrate pairwise distance
4: **for** $i \in \mathcal{R}$ **do**                 ▷ Select the first token
5:   $d_{\min}[i] \leftarrow \min_{j \in \mathcal{R}, j \neq i} D_{i,j}^{(f)}$
6: **end for**
7: $k \leftarrow \arg\max d_{\min}$
8: Move $k$ from $\mathcal{R}$ to $\mathcal{I}^{(f)}$
9: **while** $|\mathcal{I}^{(f)}| < \tilde{M}$ **do**            ▷ Iteratively add the subsequent tokens
10:   Initialize $d_{\min} \leftarrow \inf$
11:   **for** $i \in \mathcal{R}$ **do**
12:    $d_{\min}[i] \leftarrow \min_{j \in \mathcal{I}^{(f)}} D_{i,j}^{(f)}$
13:   **end for**
14:   $k \leftarrow \arg\max d_{\min}$
15:   Move $k$ from $\mathcal{R}$ to $\mathcal{I}^{(f)}$
16: **end while**
17: **return** $\mathcal{I}^{(f)}$

---

a Max-Min Diversity Problem (MMDP), calibrated by [CLS] attention and event relevance. The detailed implementation is provided in Alg. 3.

**Inner-LLM Pruning.** As illustrated in Sec. 2.1, the hybrid compression framework balances efficiency and performance, which preserves sufficient visual information input to the LLM, preventing the loss of important information. FlashVID employs this design for better performance, which retains more visual tokens before the LLM and prunes at a relatively high layer. We set the pruning layer $K = 20$ and the expansion factor $f_e = 1.25$ without careful tuning for LLaVA-OneVision, LLaVA-Video, and Qwen2.5-VL.

## C.3 TOKEN BUDGET ALIGNMENT

To ensure a fair comparison, we employ a simple and effective strategy that aligns the average number of visual tokens processed by each Transformer layer to meet a similar computational cost, following (Shao et al., 2025b). Eq. 3 presents the Floating Point Operations (FLOPs) formula of the standard Transformer architecture for generality. In this paper, we evaluate three representative VLLMs: LLaVA-OneVision (Li et al., 2025a), LLaVA-Video (Zhang et al., 2024), Qwen2.5-VL (Bai et al., 2025b), which share similar LLM architectures that employ Group Query Attention (Ainslie et al., 2023) and SwiGLU (Dauphin et al., 2017) non-linear activation. The computational FLOPs of these three LLMs can be formulated as:

$$\text{FLOPs} = L \times (2nd^2(1 + g/h) + 2n^2 d + 3ndm), \tag{12}$$

where $n$ is sequence length, $d$ the hidden dimension, $m$ the intermediate dimension of FFNs, $g$ the number of key/value heads, and $h$ the number of attention heads. Since the number of visual tokens $n_v$ dominates the sequence length $n$, the sequence length $n$ can be approximated by $n_v$.

To clarify how visual token numbers are determined at each stage. We provide a detailed explanation. Let $\bar{R}$ be the average retained visual tokens per Transformer layer, $M$ be the number of tokens entering the LLM (after before-LLM compression), $K$ be the pruning layer index, $L$ be the number of Transformer layers in LLM, and $R$ be the number of retained visual tokens (after inner-LLM pruning). Then we have the following equation.

$$\bar{R}L = MK + R(L - K). \tag{13}$$

We introduces an expansion factor $f_e$ such that $M = f_e \bar{R}$ ; thus, we have:

$$R = \frac{\bar{R}(L - f_e K)}{L - K}. \tag{14}$$

And the inner-LLM pruning ratio $r$ becomes:

$$r = \frac{R}{M} = \frac{L - f_e K}{f_e(L - K)}. \tag{15}$$

Such a simple token budget alignment strategy enables fair comparisons between different acceleration frameworks.

## D    RELATED WORK

**Multimodal Large Language Models.**    Recent advances in deep learning (He et al., 2016; Vaswani et al., 2017; Devlin et al., 2019; Dosovitskiy et al., 2021; Radford et al., 2021; He et al., 2022; Cui et al., 2022; 2023; Peng et al., 2024a; Yang et al., 2024c; Wang et al., 2024a) have benefited traditional computer vision tasks, such as semantic segmentation and object detection (Tian et al., 2020; Lai et al., 2021; Jiang et al., 2021; Peng et al., 2023; Tian et al., 2022b; Luo et al., 2023; Peng et al., 2024b; Tian et al., 2022a; 2019; 2023; Ning et al., 2023; Shao et al., 2024; Wang et al., 2025a;b). In particular, transformer-based architectures and large-scale pretraining have increasingly driven the success of Large Language Models (LLMs) (Touvron et al., 2023; Grattafiori et al., 2024; Yang et al., 2024a;b; 2025a; Lai et al., 2024b; Peng et al., 2025; Liu et al., 2024a; Guo et al., 2025), exhibiting strong generalization and reasoning capabilities. Building upon LLMs, Multimodal Large Language Models (MLLMs) (Liu et al., 2023; 2024b;c; Dai et al., 2023; Li et al., 2023; Comanici et al., 2025; Bai et al., 2025b;a; Wang et al., 2025c; Li et al., 2025c) extend the input modality from text to multimodalities (such as image, audio, and video) by coupling modality encoders with LLM backbones. So far, MLLMs have revolutionized traditional computer vision tasks. For example, representative works like LISA (Lai et al., 2024a) and LISA++ (Yang et al., 2023) study reasoning segmentation powered by MLLMs.

**Video Large Language Models.**    With the rapid advancement of MLLMs, Video Large Language Models (VLLMs) (Li et al., 2025a; Zhang et al., 2024; Bai et al., 2025b;a; Shen et al., 2024; Li et al., 2024b; Maaz et al., 2024; Comanici et al., 2025) have gained increasing attention. Mainstream VLLMs directly process raw video tokens with an optional pooling operation. LLaVA-OneVision (Li et al., 2025a) demonstrates strong video understanding capabilities through task transfer from images. To achieve fine-grained spatiotemporal modeling, some VLLMs employ elaborate designs. LLaVA-Video (Zhang et al., 2024) introduces newline tokens to distinguish spatiotemporal positions. Qwen2-VL (Wang et al., 2024b) and Qwen2.5-VL (Bai et al., 2025b) use Multimodal Rotary Position Embedding (MRoPE). Qwen3-VL (Bai et al., 2025a) employs the Deepstack mechanism (Meng et al., 2024), which extracts visual tokens from intermediate layers of the visual encoder and injects them into the LLM, preserving rich visual information.

To achieve a comprehensive evaluation, we evaluate our method on three representative VLLMs (i.e., LLaVA-OneVision, LLaVA-Video, and Qwen2.5-VL) with significantly different architectures and characteristics.

**Visual Token Compression.**    Token compression has emerged as an effective technique that reduces computational complexity in transformer architectures, such as Vision Transformers (ViTs) (Dosovitskiy et al., 2021) and Large Language Models (LLMs). ToMe (Bolya et al., 2023) gradually merges similar tokens in ViTs. FastV (Chen et al., 2024) identifies text-relevant visual tokens based on text-to-visual attention in the LLM. PyaramidDrop (Xing et al., 2024) and SparseVLM (Zhang et al., 2025e) progressively prunes visual tokens. VisionZip (Yang et al., 2025c), LLaVA-PruMerge (Shang et al., 2025), and VisPruner (Zhang et al., 2025c) filter salient visual tokens via [CLS] attention, while DivPrune (Alvar et al., 2025) selects based on diversity. TopV (Yang et al., 2025b) formulates token selection as an optimization problem. VScan (Zhang et al., 2025a) combines global and local scans for informative visual token selection.

However, the above methods only focus on spatial redundancy compression. To address this, several token compression frameworks for VLLMs have been proposed. DyCoke (Tao et al., 2025) merges

redundant tokens in each segment. PruneVID (Huang et al., 2025) distinguishes static and dynamic tokens. STTM (Hyun et al., 2025) models video redundancy by a quadtree. FrameFusion (Fu et al., 2025b) performs both merging and pruning in the LLM. HoliTom (Shao et al., 2025a) combines global redundancy-aware video partition with spatial and inner-LLM compression. FastVID (Shen et al., 2025) employs density-based token pruning. DyTok (Li et al., 2025b) dynamically allocates token budget to each frame or segment, serving as a plug-and-play module for existing token compression methods.

# E  MORE VISUALIZATIONS

## E.1  TREE-BASED SPATIOTEMPORAL TOKEN MERGING

Due to the dynamic and evolving nature of video, the most semantically correlated visual features in adjacent frames are likely to experience variation in position, scale, orientation, and other attributes over time. To address this challenge, we propose the Tree-based Spatiotemporal Token Merging (TSTM) mechanism, which models video redundancy by spatiotemporal redundancy trees in a unified way. It enables capturing fine-grained video dynamics. Fig. 5 presents more visualizations of TSTM, highlighting the unique advantages of our TSTM for better spatiotemporal redundancy compression.

## E.2  QUALITATIVE ANALYSIS ON LLAVA-ONEVISION

As illustrated in Tab. 1, we evaluate our FlashVID on LLaVA-OneVision at $R \in \{25\%, 20\%, 15\%, 10\%\}$. Notably, at higher retention ratios (i.e., $R = 25\%, 20\%, 15\%$), FlashVID surpasses the vanilla LLaVA-OneVision, indicating a "*less is more*" pattern where excessively redundant tokens may degrade performance. Additionally, FlashVID preserves performance of **99.1%** under extreme compression (e.g., $R = 10\%$). Fig. 6 presents four qualitative examples comparing LLaVA-OneVision with and without FlashVID, which indicates FlashVID enables fine-grained spatiotemporal redundancy compression, providing compact yet informative video representation.

## E.3  QUALITATIVE ANALYSIS ON QWEN2.5-VL

In this work, we explore extending VLLMs to process more video frames under a fixed computational budget through visual token compression. As reported in Tab. 3 and Tab. 8, VLLMs benefit from longer temporal context. Fig. 7 presents four qualitative examples comparing Qwen2.5-VL with and without FlashVID, highlighting its ability to capture richer temporal information. FlashVID enables Qwen2.5-VL (Bai et al., 2025b) to process **10**× more frames (160 vs. 16) within the same computational cost, providing compact yet informative video representations and improving the model performance.

## E.4  VISUALIZATIONS OF ADTS

As shown in fig. 8, we compare token selection results by ADTS with and without event relevance calibration. Event relevance calibration helps identify the key visual tokens, thereby improving the performance of those tasks requiring fine-grained understanding.

## E.5  VISUALIZATIONS OF FAILURE CASES IN TSTM

As illustrated in Fig. 9, we present visualizations of failure cases in our Tree-based Spatiotemporal Token Merging (TSTM). Although TSTM enables fine-grained spatiotemporal redundancy compression, it might result in merging operations with semantic confusion such as merging tokens from different entities with similar semantic information.

## E.6  VISUAL PERCEPTION LAYERS

We empirically found that certain transformer layers (deep layers) of VLLMs possess strong visual perception capabilities. These visual perception layers can typically identify keyframes. Fig. 10

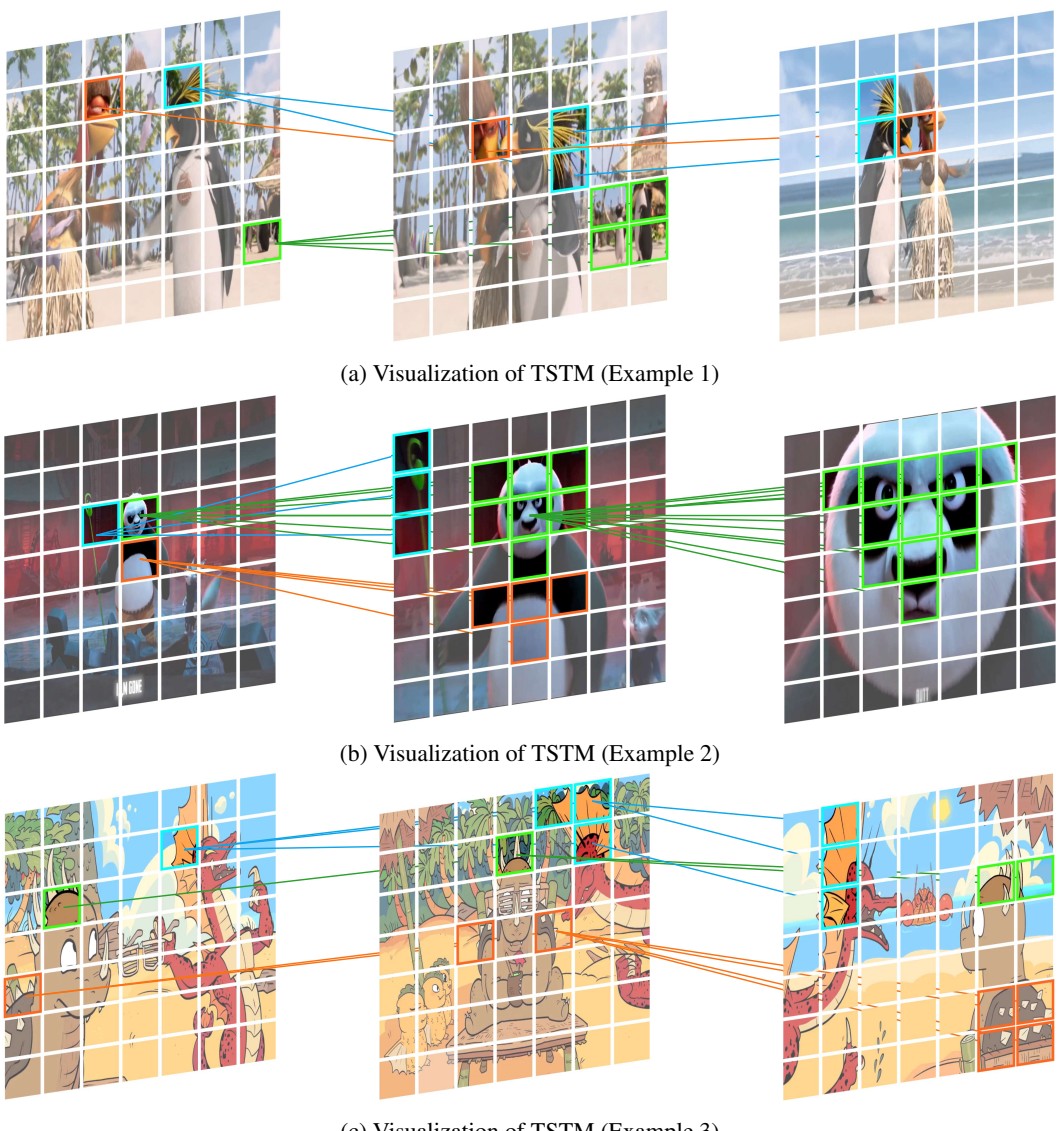

(a) Visualization of TSTM (Example 1)

(b) Visualization of TSTM (Example 2)

(c) Visualization of TSTM (Example 3)

Figure 5: **Visualizations of Tree-based Spatiotemporal Token Merging (TSTM).** We select three consecutive video frames that show obvious variations in spatial locations, scale, and orientation for each case to illustrate the advantages of our TSTM in FlashVID. TSTM jointly models spatial and temporal redundancy via spatiotemporal redundancy trees for capturing fine-grained spatiotemporal relationships; thus, it achieves better spatiotemporal redundancy compression.

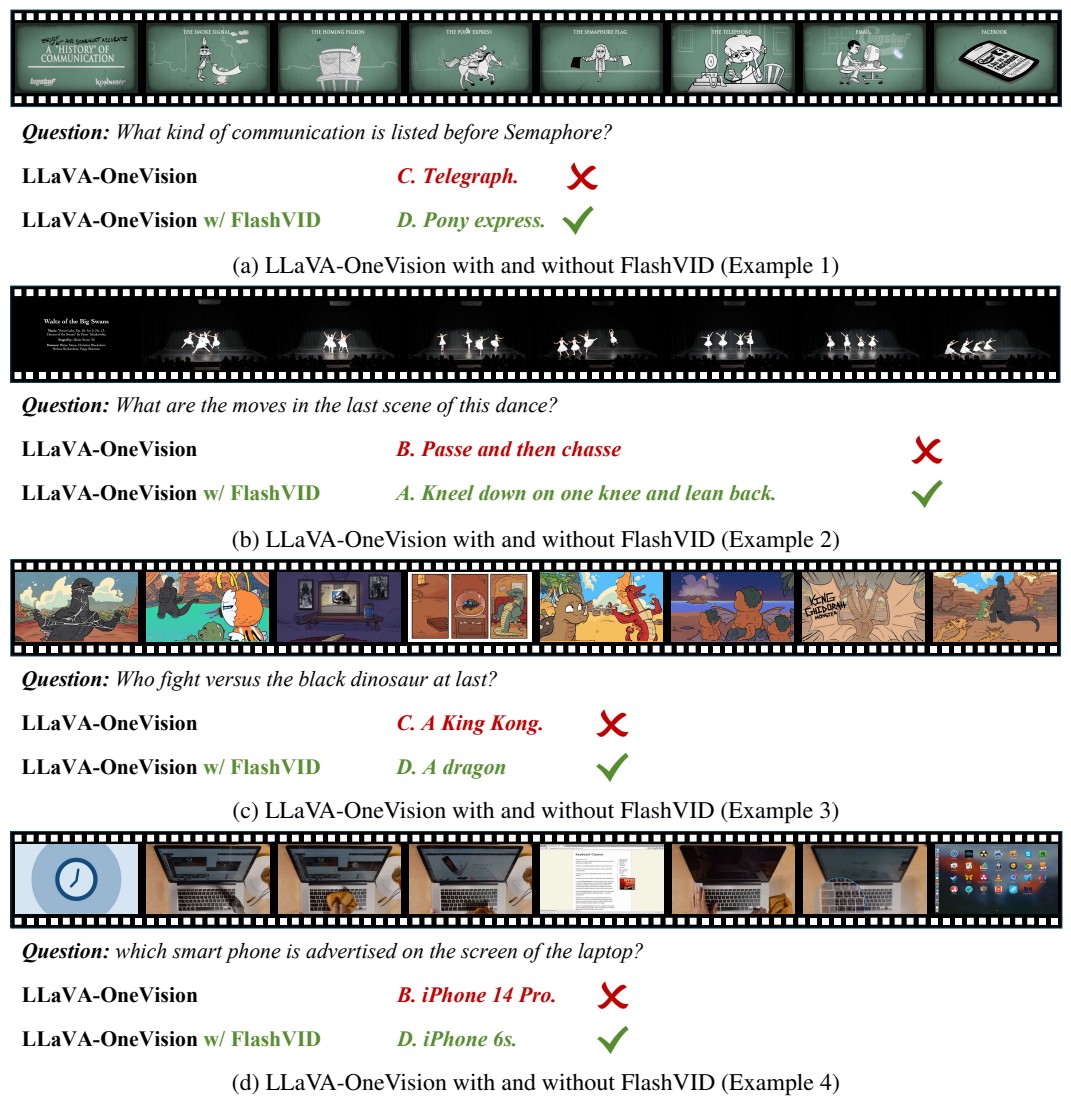

(a) LLaVA-OneVision with and without FlashVID (Example 1)

(b) LLaVA-OneVision with and without FlashVID (Example 2)

(c) LLaVA-OneVision with and without FlashVID (Example 3)

(d) LLaVA-OneVision with and without FlashVID (Example 4)

Figure 6: **Qualitative comparison of LLaVA-OneVision with and without FlashVID.** We conduct qualitative analysis on LLaVA OneVision with and without FlashVID under a 25% retention ratio. We observe an interesting phenomenon: in some examples, LLaVA OneVision with FlashVID compression can answer the questions correctly, whereas the original model with full visual tokens input gives incorrect answers, unveiling a *"less-is-more"* pattern where excessive visual tokens input may degrade model performance.

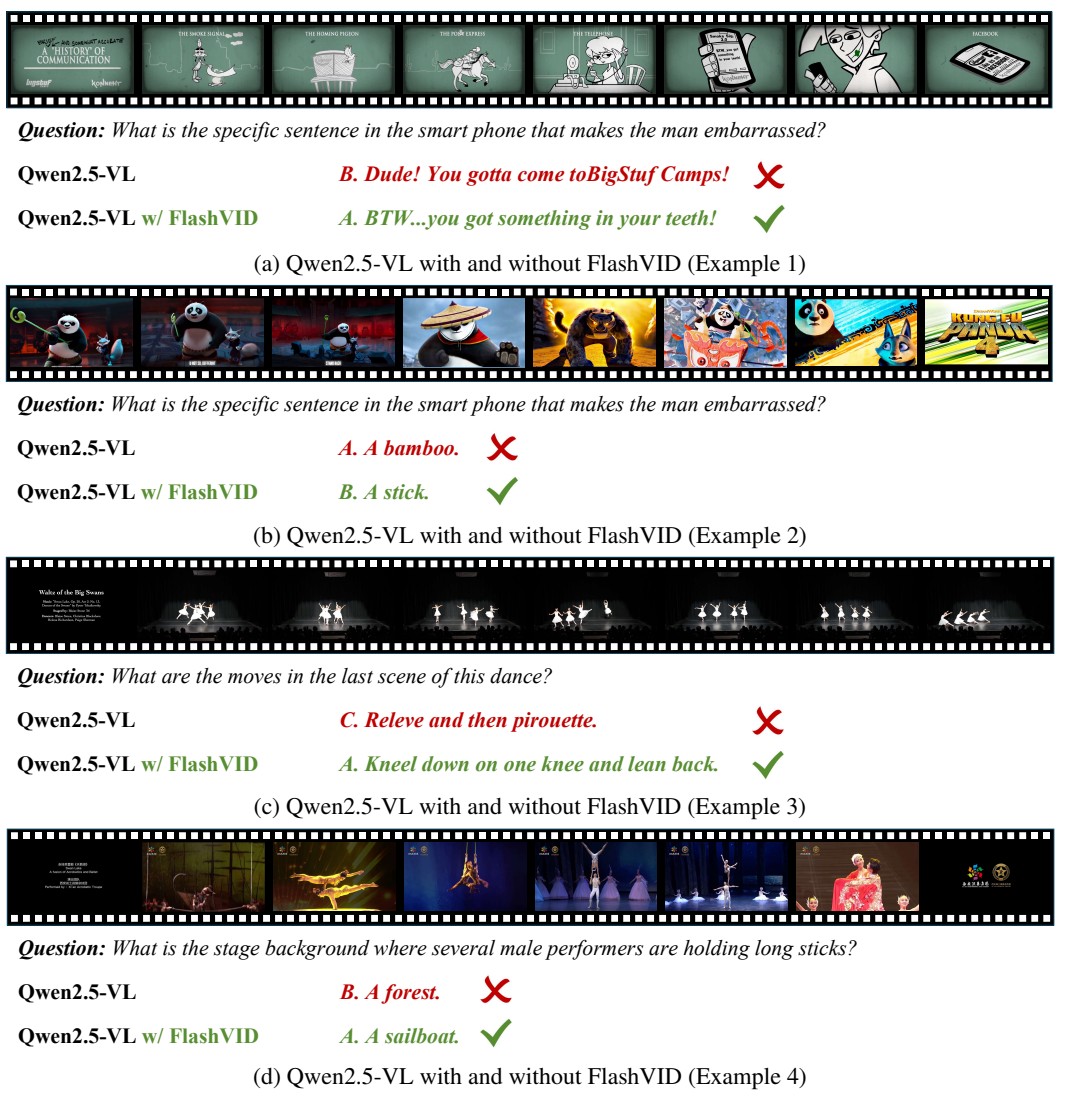

Figure 7: **Qualitative comparison of Qwen2.5-VL with and without FlashVID.** The vanilla model processes only 16 sampled frames, which limits its ability to capture sufficient temporal information. In contrast, Qwen2.5-VL can handle 160 (**10×**) frames with FlashVID while maintaining the overall computational budget, yielding more accurate predictions by leveraging longer temporal context.

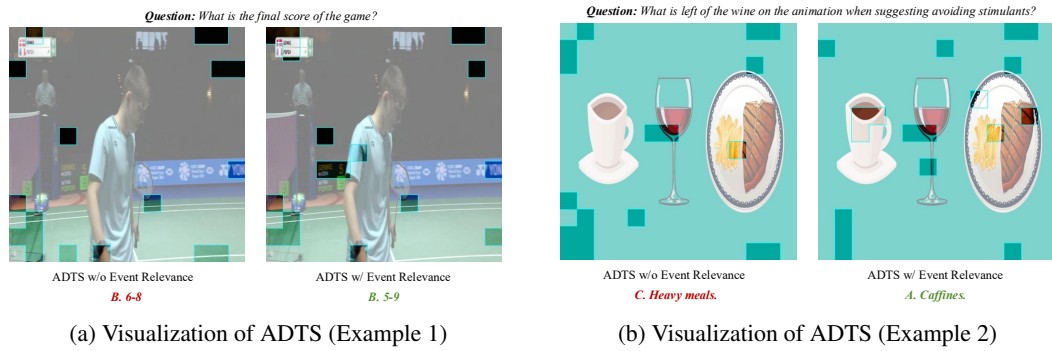

Figure 8: **Comparisons of ADTS with and without event relevance calibration.** ADTS employs event relevance calibration terms to identify the tokens most relevant to the video event.

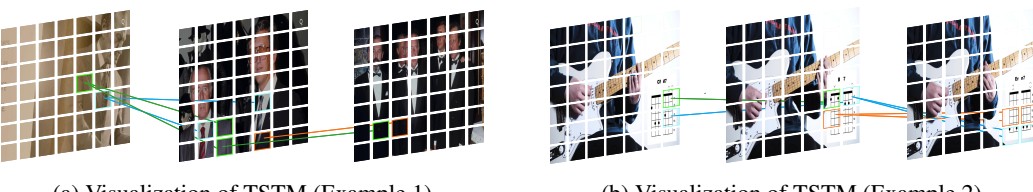

(a) Visualization of TSTM (Example 1)          (b) Visualization of TSTM (Example 2)

Figure 9: **Visualizations of failure cases in TSTM.**

presents several visualizations of visual perception layers. Building upon this insight, we hypothesize that token compression at these layers yields negligible performance degradation.

To balance efficiency and performance, FlashVID adopts a hybrid compression paradigm that retains more visual tokens and prunes visual tokens in the LLM to control the overall computational budget. Hence, we consistently set the pruning layer $K = 20$ (a relatively high layer for LLaVA-OneVision, LLaVA-Video, and Qwen2.5-VL at 7B scale) without careful tuning.

## F    USAGE OF LARGE LANGUAGE MODELS

In this work, Large Language Models (LLMs) are only used for polishing the paper writing. They are not involved in research ideation, experimental design, data analysis, or the formulation of conclusions. All substantive intellectual contributions are made by the authors.

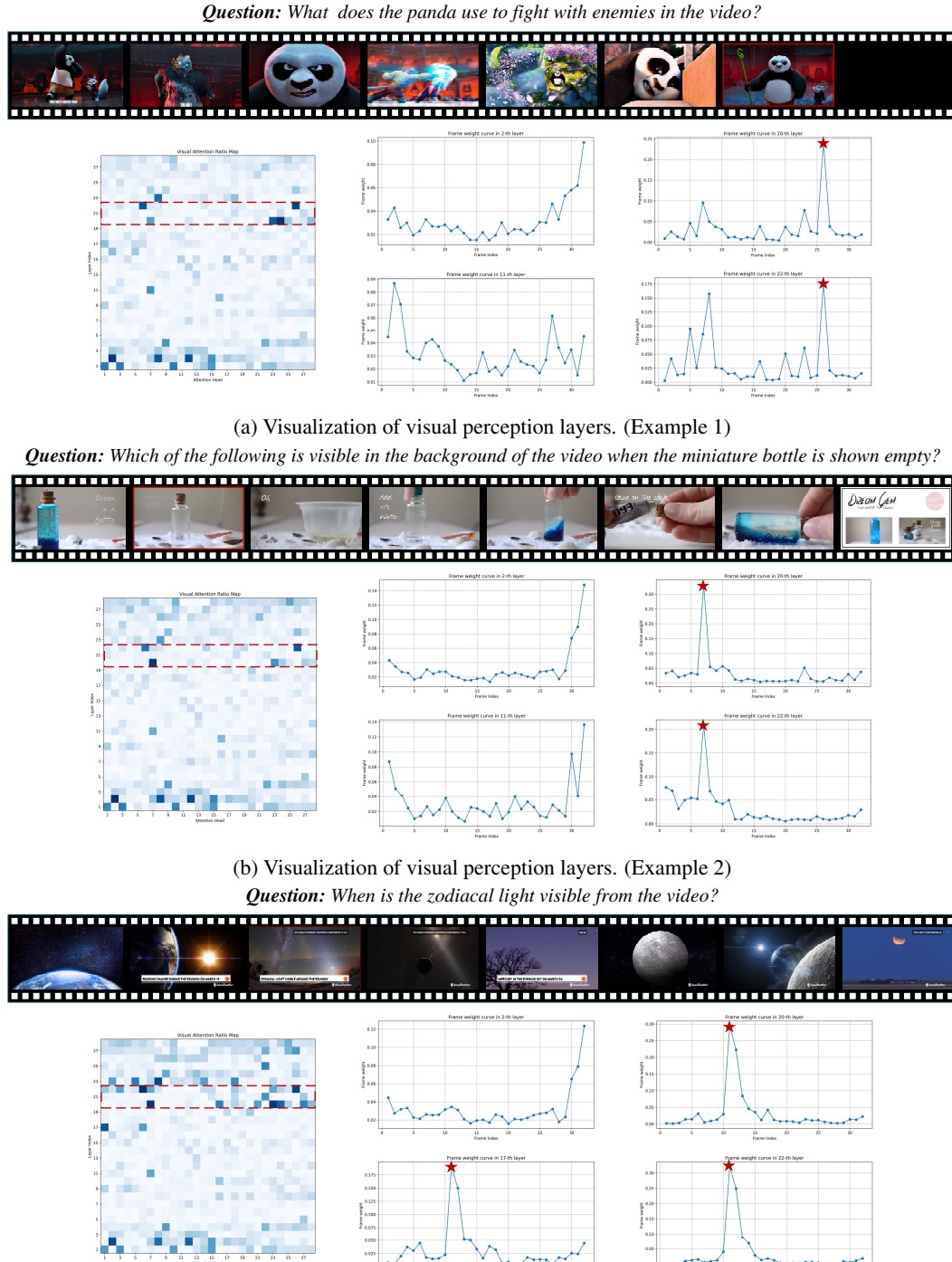

Figure 10: **Visualizations of visual perception layers.** We empirically found that certain layers in VLLMs have strong visual perception capabilities, which can accurately recognize keyframes. We hypothesize that pruning visual tokens guided by attention weights in these layers could filter tokens most relevant to the text query, achieving a better pruning performance.

