# OpenReview forum: "FlashVID: Efficient Video Large Language Models via Training-free Tree-based Spatiotemporal Token Merging"
_ICLR.cc/2026/Conference — ICLR 2026 Oral_

### Official Review · Reviewer_nJgS · 2025-11-01

[review text omitted: it was posted to a different submission]

---

> ### Author Response · Authors · 2025-11-18
> **Inequiry for the Mismatched Review for the Submission Paper 5848**
>
> Dear Reviewer nJgS,
>
> Thank you for your time and effort in reviewing our submission.
>
> After carefully reviewing the feedback, we found it difficult to align the feedback with the proposed method in our paper. This leads us to suspect that there may have been a mix-up with another submission.
>
> Specifically, the feedback underlines three key innovations: **"Flash Temporal Attention (FTA)"**, **"Segment-Level Progressive Compression (SPC)"**, and **"Streaming Memory Replay (SMR)"**. However, our paper **5848** proposes a distinct framework that adopts **"Tree-based Spatiotemporal Token Merging (TSTM)"** mechanism for fine-grained spatiotemporal redundancy compression and the **"Attention and Diversity-based Token Selection (ADTS)"** module for frame-wise representative token selection. Given this mismatch, we are unable to address the concerns and questions raised regarding the SPC and SMR modules, even though the same name "FlashVID" has been mentioned.
>
> Therefore, we would greatly appreciate it if you could double-check whether the review was assigned to the correct submission. We are still looking forward to your valuable insights and feedback.
>
> Thank you again for your understanding and consideration.
>
> Best regards,
>
> The Authors of Submission #5848

---

> > ### Comment · Reviewer_nJgS · 2025-11-18
> > **Correct Review**
> >
> > Dear Authors,
> >
> > Thank you for your message, and I sincerely apologize for the confusion. The feedback you received earlier was mistakenly attached and did not correspond to your submission.
> >
> > I have now corrected the review and ensured that the proper comments for your paper are in place. Please kindly refer to the updated review.
> >
> > My apologies again for the oversight, and thank you for your understanding.
> >
> > Best regards,
> > Reviewer nJgS

---

> ### Author Response · Authors · 2025-11-21
> **Response to Weakness 1 (Benchmark computational complexity)**
>
> > **Weakness 1**: It would be helpful to benchmark the compute complexity of TSTM, especially when the high-resolution vision encoders are employed.
>
> Thank you for your insightful suggestion.
>
> It is worth noting that FlashVID can enable Video Large Language Models (VLLMs) to achieve a satisfactory speedup with negligible performance drop, though it consists of several steps (i.e., video partition, ADTS, TSTM, inner-LLM pruning).
>
> As illustrated in **Appendix A.3.1**,  we've conducted an efficiency experiment on LLaVA-Video (see **Table 16** in our revision). Our FlashVID can speed up **5.3x** prefilling time and **1.9x** Time-To-First-Token (TTFT), compared to the vanilla LLaVA-Video. Following FastVID, the prefilling time contains token compression and the first LLM forward time, while the TTFT includes extra vision encoding time.
>
> To address your concern, we further conduct efficiency experiments on LLaVA-OneVision using a single NVIDIA A100 GPU, comparing to FastVID on VideoMME. As shown in the following table, we compare the prefilling time and TTFT speedups with FastVID under the same performance.
>
> | Method          | Retention Ratio (%) / FLOPs (T) | GPU Memory (GB) | Vision Encoding (ms) | Compression (ms) | LLM Forward (ms) | Prefilling Time (ms) | TTFT (ms)        | Rel. Acc. (%) |
> | --------------- | ------------------------------- | --------------- | -------------------- | ---------------- | ---------------- | -------------------- | ---------------- | ------------- |
> | LLaVA-OneVision | 100.0 / 113.4                   | 30.5            | 785.0                | -                | 1220.8           | 1220.8  (1.0x)       | 2005.8 (1.0x)    | 100.0         |
> | FastVID         | 25.0 / 22.4                     | 27.0            | 785.0                | 28.6             | 273.2            | 301.8 (4.0x)         | 1086.8 (1.8x)    | 99.5          |
> | **FlashVID**    | 10.0 / 8.6                      | 26.9            | 785.0                | 60.2             | 133.1            | 193.3 (**6.3x**)     | 978.3 (**2.1x**) | **99.7**      |
>
> Remarkably, FlashVID preserves **99.7%** relative performance to vanilla LLaVA-OneVision under a **10%** retention ratio, while FastVID achieves a similar performance at a **25%** retention ratio. Consequently, FlashVID achieves **6.3x** prefilling and **2.1x** TTFT speedup, largely outperforming FastVID.
>
> We hope the above efficiency analysis can address your concern regarding the efficiency of our FlashVID.

---

> ### Author Response · Authors · 2025-11-21
> **Response to Weakness 2 (Sensitivity to multiple hyperparameters)**
>
> > Dependence on hyper-parameters seems non-trivial. The work introduces several hyper-parameters, it is unclear whether these hyper-parameters transfer across models or need tuning per VLLM or per real-world task. Does a single set of hyper-parameters transfer to datasets with drastically different appearances (e.g., sports videos, driving videos, egocentric videos)?
>
> Thank you for pointing out this concern.
>
> Although FlashVID incorporates several hyperparameters, we ensure **consistent** hyperparameter settings across different Video Large Language Models (VLLMs) and video understanding benchmarks in all our experiments, showcasing the robustness of our method. Since our video partition module is built upon DySeg[1], we strictly follow the same hyperparameter settings, in which the segment threshold $S_\tau$  and the number of minimum segments $M_s$ are set to $0.9$ and $8$ (See **Appendix C.2** in our revision), respectively. Other hyperparameters in FlashVID (e.g., temporal merging threshold $T_\tau$, token distribution $\alpha$, and expansion factor $f_e$) are set to the same values in all experiments (see **Section 4.1** and **Appendix C.2**).
>
> Extensive experiments (see **Section 4.2, Appendix A**) conducted on three representative VLLMs (i.e., LLaVA-OneVision, LLaVA-Video, Qwen2.5-VL) and five widely-used video understanding benchmarks (i.e., VideoMME, EgoSchema, LongVideoBench, MVBench, MLVU) under a range of retention ratios using the **same** set of hyperparameters demonstrate that FlashVID significantly outperforms previous state-of-the-art acceleration frameworks (e.g., FastV, VisionZip, FastVID). Such consistent and superior performance highlights the robustness and generalizability of our approach.
>
> To summarize, we employ the same hyperparameter settings in all our experiments, in which FlashVID consistently outperforms previous methods across all settings, demonstrating the superior robustness and generalizability of our method.
>
> We hope the above analysis can address your concern regarding the robustness of our method.

---

> ### Author Response · Authors · 2025-11-21
> **Response to Weakness 3 (Visualization of action-heavy clips in TSTM)**
>
> > **Weakness 3**: While some TSTM trees are visualized, it’s still difficult to intuitively understand what kinds of tokens get merged or preserved, especially in rapidly changing scenes. It would be helpful to show some examples (e.g., merging patterns in action-heavy clips) to visualize the merging behavior.
>
> Thank you for your comment.
>
> As illustrated in **Algorithm 1 (line 9-18)**, our Tree-based Spatiotemporal Token Merging (TSTM) module first constructs spatiotemporal redundancy trees based on feature similarities between adjacent frames, where each visual token in current frame will be connected to its anchor token in the previous frame with the largest cosine similarity if the similarity exceeds a temporal merging threshold $T_\tau$. After building the spatiotemporal redundancy trees, each tree is aggregated into a single token representation via feature averaging.
>
> A conceptual visualization of TSTM is presented in **Figure 3(c)**, while several typical visualizations of TSTM are provided **in Figure 5 in Appendix E.1**. The qualitative visualizations show a unique advantage of TSTM, which flexibly captures fine-grained spatiotemporal relationships. Intuitively, tokens in adjacent frames with similar semantics will be merged; otherwise will be persevered. We are still working on visualizing the merging patterns on action-heavy clips. Unfortunately, we can not directly present visualizations here. Hence, we will add additional visualizations of TSTM **in Appendix E.1** in the revision, showing the merging patterns in rapidly changing scenes.
>
> We hope the extensive visualizations of TSTM can help intuitively understand the merging patterns in TSTM. We appreciate your feedback and are happy to address any further suggestions or concerns you may have.

---

> ### Author Response · Authors · 2025-11-21
> **Response to Question 1 (Visualization of failure cases in TSTM)**
>
> > **Question 1**: It would be informative to show examples where FlashVID fails: e.g., where merging induces semantic confusion.
>
> Thank you for your insightful comment.
>
> Although our Tree-based Spatiotemporal Token Merging (TSTM) enables fine-grained spatiotemporal redundancy compression and achieves superior performance (see **Table 1-3, Table 6-8**), it might result in merging operations with semantic confusion. Unfortunately, we can not directly present visualizations here. Hence, some visualizations of failure cases in TSTM are added in **Figure 6 in Appendix E.1** in the revision. These examples show that TSTM still faces challenges such as merging tokens from different entities with similar semantic information. Such merging may cause semantic confusion, potentially undermining those tasks requiring fine-grained understanding.

---

> ### Author Response · Authors · 2025-11-27
> **Hoping for a Discussion on Our Rebuttal for Paper 5848**
>
> Dear Reviewer,
>
> I hope this message finds you well. We have carefully responded to the questions and concerns you previously raised, and we would greatly appreciate any further feedback you might be able to provide. Your insights are highly valuable to us, and we remain fully available to clarify or refine any remaining points.
>
> As the discussion deadline approaches, we would appreciate your response at your earliest convenience, as it would greatly help us move the discussion forward.
>
> Thank you sincerely for your time and effort in reviewing our paper.
>
> Best regards,
>
> The Authors

---

> > ### Comment · Reviewer_nJgS · 2025-11-28
> >
> > Thanks authors for the detailed response. It addressed my concerns. Therefore, I would like to raise my rating to "6" and have a positive recommendation of this submission.

---

> > > ### Author Response · Authors · 2025-11-28
> > > **Thank you for your positive feedback!**
> > >
> > > Dear Reviewer nJgS,
> > >
> > > Thank you very much for your time and effort in reviewing our paper and providing insightful comments. We are sincerely grateful for your positive feedback on our revisions and your willingness to reconsider the score.
> > >
> > > We are pleased to hear that our responses and the manuscript modifications have successfully addressed your concerns, particularly regarding the sensitivity of the hyperparameters in our approach, the efficiency of our FlashVID, and the inclusion of additional visualizations of our TSTM module.
> > >
> > > Your insightful comments help us improve the quality and clarity of our paper. Once again, we appreciate your guidance and support for our work.
> > >
> > > Best regards,
> > >
> > > The authors

---

### Official Review · Reviewer_tpdQ · 2025-11-01

**Soundness:** 4
**Presentation:** 4
**Contribution:** 3
**Rating:** 4
**Confidence:** 5

**Summary:**

This paper proposes a training-free inference acceleration framework called FlashVID, aimed at addressing the high computational cost issues caused by processing massive visual tokens in video large language models (VLLMs). FlashVID employs a two-stage compression strategy: first, it selects representative tokens within each frame using the "Attention and Diversity-based Token Selection (ADTS)" module; second, it constructs a "spatio-temporal redundancy tree" across frames through the "Tree-based Spatio-Temporal Token Merging (TSTM)" mechanism to jointly model and merge similar tokens that change dynamically over time. Experiments were validated on three VLLMs and five video benchmarks, maintaining 99.1% of LLaVA-OneVision's performance while retaining only 10% of the tokens, and achieving a 10x increase in the number of frames processed by Qwen2.5-VL under the same computational budget, resulting in an 8.6% improvement.

**Strengths:**

1. FlashVID is a training-free framework, can be used as a plug-and-play module applied to existing trained VLLMs without expensive training costs.
2. Addressed a key pain point of previous VLLM acceleration methods: they typically compress spatial and temporal redundancies independently, or rely on a single spatial correspondence for temporal merging. TSTM can flexibly track and merge similar tokens that change dynamically over time in terms of spatial location, scale, or direction by building a "spatio-temporal redundancy tree," which better aligns with the dynamic characteristics of videos.

**Weaknesses:**

1. This method requires multiple hyperparameters that need empirical tuning, which may affect its plug-and-play performance across different models or datasets. For example $T_{\tau}$, $f_{e}$, $\alpha$。
2. The paper mentions the interesting phenomenon of "less is more" and attributes it to "excessive visual token input may introduce noise," which is a reasonable assumption but lacks more in-depth quantitative or qualitative analysis to clarify how these "noise" specifically affect VLLM's attention or internal representations. Moreover, this phenomenon was only tested on the llava-onevision model, making the argument less persuasive. If this phenomenon can also be observed on models like qwen-vl and internvl, it would further strengthen the argument.
3. ADTS introduced an "event relevance" to calibrate "importance", but it uses global average pooling (GAP) to average the entire frame information into a single vector. If the key event in a video is a local small action and most of the area is static background, the GAP vector will mainly represent the background, which may lead ADTS to mistakenly consider this key local event as "irrelevant" and possibly discard it. Is this global heuristic method a significant weakness for tasks requiring fine-grained understanding?

**Questions:**

1. "We set K = 20 for LLaVA-OneVision, LLaVA-Video, and Qwen2.5-VL." Has K=20 been analyzed? Or is it set based on experience, and if it's a new model, would this configuration also apply?

---

> ### Author Response · Authors · 2025-11-21
> **Response to Weakness 1 (Sensitivity to multiple hyperparameters)**
>
> > **Weakness 1**: This method requires multiple hyperparameters that need empirical tuning, which may affect its plug-and-play performance across different models or datasets. For example $T_\tau$, $f_e$, $\alpha$.
>
> Thank you for pointing out this concern.
>
> Although FlashVID incorporates several hyperparameters, we ensure **consistent** hyperparameter settings across different Video Large Language Models (VLLMs) and video understanding benchmarks in all our experiments, showcasing the robustness of our method. Since our video partition module is built upon DySeg[1], we strictly follow the same hyperparameter settings, in which the segment threshold $S_\tau$  and the number of minimum segments $M_s$ are set to $0.9$ and $8$ (See **Appendix C.2** in the revision), respectively. Other hyperparameters in FlashVID (e.g., temporal merging threshold $T_\tau$, token distribution $\alpha$, and expansion factor $f_e$) are set to the same values in all experiments (see **Section 4.1** and **Appendix C.2**).
>
> Extensive experiments (see **Section 4.2, Appendix A**) conducted using the **same** set of hyperparameters on three representative VLLMs (i.e., LLaVA-OneVision, LLaVA-Video, Qwen2.5-VL) and five widely-used video understanding benchmarks (i.e., VideoMME, EgoSchema, LongVideoBench, MVBench, MLVU) under a range of retention ratios demonstrate that FlashVID significantly outperforms previous state-of-the-art acceleration frameworks (e.g., FastV, VisionZip, FastVID). Such consistent and superior performance highlights the robustness and generalizability of our approach.
>
> To summarize, we employ the same hyperparameter settings in all our experiments, in which FlashVID consistently outperforms previous methods across all settings, demonstrating the superior robustness and generalizability of our method. We hope the above analysis can address your concern regarding the robustness of our method.
>
> [1] FastVID: Dynamic Density Pruning for Fast Video Large Language Models (NeurIPS 2025)

---

> ### Author Response · Authors · 2025-11-21
> **Response to Weakness 2 (“Less is more” phenomenon lacks deeper analysis)**
>
> > **Weakness 2**: The paper mentions the interesting phenomenon of "less is more" and attributes it to "excessive visual token input may introduce noise," which is a reasonable assumption but lacks more in-depth quantitative or qualitative analysis to clarify how these "noise" specifically affect VLLM's attention or internal representations. Moreover, this phenomenon was only tested on the llava-onevision model, making the argument less persuasive. If this phenomenon can also be observed on models like qwen-vl and internvl, it would further strengthen the argument.
>
> Thank you for your insightful comments.
>
> As illustrated in Table 1, our FlashVID outperforms the vanilla LLaVA-OneVision at 15%, 20%, and 25% retention ratios. However, we didn't observe the same trend on LLaVA-Video and Qwen2.5-VL. Therefore, we revise the "less is more" pattern in Appendix E.4 in the revision. Besides, we would like to clarify that this does not affect the efficacy of our primary contribution, as FlashVID enables efficient VLLMs with negligible performance drop (see **Table 1-3, Table 6-8**).
>
> We sincerely appreciate your feedback and are happy to address any further suggestions or concerns you may have.

---

> ### Author Response · Authors · 2025-11-21
> **Response to Weakness 3 (Potential limitation of event relevance in ADTS)**
>
> > **Weakness 3**: ADTS introduced an "event relevance" to calibrate "importance", but it uses global average pooling (GAP) to average the entire frame information into a single vector. If the key event in a video is a local small action and most of the area is static background, the GAP vector will mainly represent the background, which may lead ADTS to mistakenly consider this key local event as "irrelevant" and possibly discard it. Is this global heuristic method a significant weakness for tasks requiring fine-grained understanding?
>
> Thank you for your insightful comments.
>
> Event relevance is one of the calibration terms in our Attention and Diversity-based Token Selection (ADTS) module, which is designed to identify those visual tokens most relevant to the video event, while the [CLS] attention calibration term highlights the informative tokens in each frame.
>
> To address your concern, we further conduct ablation studies on calibration terms in ADTS on LLaVA-OneVision, shown in the following table. The experimental results indicate that both [CLS] attention and event relevance calibrations bring performance gains, and the peak performance is achieved when combined with two calibration terms. Notably, these new results have been added in **Appendix A.3** in our revision (see **Table 14**).
>
> |Method|VideoMME|EgoSchema|LongVideoBench|MVBench|Rel. Acc.|
> |---|---|---|---|---|---|
> |DTS|55.7|**60.3**|55.3|55.5|97.1|
> |w/ [CLS] attention|56.0|60.2|55.1|56.8|97.6|
> |w/ Event Relevance|57.3|59.7|55.7|57.3|98.5|
> |ADTS|**57.8**|60.0|**56.5**|**57.4**|**99.1**|
>
> To evaluate the event relevance calibration term in ADTS for those tasks requiring fine-grained understanding, we select four tasks in VideoMME that need fine-grained perception. We compare FlashVID with and without event relevance calibration term on LLaVA-OneVision under various retention ratios, shown in the following table. The results indicate that the performance on those tasks requiring fine-grained perception benefits from the event relevance calibration term in ADTS. In addition, we present several visualization examples (see **Figure 10** in the revision) to compare the token selection result of ADTS with and without event relevance.
>
> |Method|Retention Ratio (%)|Attribute Perception|Counting Problem|Object Recognition|OCR Problems|Rel. Acc (%)|
> |---|---|---|---|---|---|---|
> |LLaVA-OneVision|100|74.3|36.9|65.3|61.9|100.0|
> |FlashVID|25|**71.2**|**39.6**|65.0|**61.2**|**99.4**|
> |w/o Event Relevance|25|69.4|39.2|**65.5**|59.0|97.8|
> |FlashVID|15|**69.4**|**35.4**|**64.7**|**58.3**|**95.6**|
> |w/o Event Relevance|15|66.7|34.7|64.7|57.6|93.8|
> |FlashVID|5|**64.9**|**32.5**|**58.2**|**48.9**|**85.8**|
> |w/o Event Relevance|5|57.7|32.5|55.4|47.5|81.0|
>
> To summarize, we provide the detailed ablation studies on calibration terms in ADTS, while comparing the results on tasks requiring fine-grained perception with and without event relevance calibration. The above results indicate that event relevance doesn't undermine the performance on those tasks requiring fine-grained perception. Moreover, we present some visualizations on ADTS, comparing whether using event relevance calibration.
>
> We hope the quantitative and qualitative results can address your concern regarding the potential limitation of the calibration term in ADTS. In addition, we appreciate your feedback and are happy to address any further suggestions or concerns you may have.

---

> ### Author Response · Authors · 2025-11-21
> **Response to Question 1 (Choice of K=20 for multiple models)**
>
> > **Question 1**: "We set K = 20 for LLaVA-OneVision, LLaVA-Video, and Qwen2.5-VL." Has K=20 been analyzed? Or is it set based on experience, and if it's a new model, would this configuration also apply?
>
> Thank you for pointing this out.
>
> It might be confusing to employ the same pruning layer setting on different VLLMs. However, it is worth noting that we set $K=20$ for LLaVA-OneVision, LLaVA-Video, and Qwen2.5-VL without explicit tuning for generality, though there may exist an optimal pruning layer $K$ in different VLLMs under a fixed computational budget.
>
> To address your concern, we provide a detailed explanation. TwigVLM [1] found that the quality of cross-modal attention weights in deep layers is relatively high; hence, applying token pruning in shallow layers guided by deep-layer attention weights can achieve a better performance. Building upon this insight, we also conducted extensive qualitative analysis (see **Figure 8** in the revision). We found a similar phenomenon that certain layers (deep layers) exhibit strong visual perception capabilities, which can accurately identify the key visual tokens.
>
> Given the three VLLMs equipped with the LLM with $L=28$ Transformer layers, we choose a relatively deep layer  $K=20$ for inner-LLM pruning without careful searching. To evaluate the robustness of our FlashVID, we further conduct ablation studies on the pruning layer $K$ on LLaVA-OneVision at a 10% retention ratio under the same token budget.
>
> |K|VideoMME|Egoschema|LongVideoBench|MVBench|Rel. Acc. (%)|
> |---|---|---|---|---|---|
> |7|57.3|59.8|55.7|57.2|98.5|
> |14|56.0|59.1|54.3|57.1|96.9|
> |18|57.7|60.0|56.5|57.4|99.1|
> |19|57.9|60.0|56.6|57.4|99.3|
> |**20**|57.8|60.0|56.5|57.4|99.1|
> |21|57.8|60.0|56.3|57.4|99.1|
> |22|57.8|60.1|56.3|57.4|99.1|
>
> The above experimental results indicate that pruning at a relatively high layer (e.g., 18-22) while maintaining overall token budget causes a negligible performance drop.
>
> To summarize, we employ the same pruning layer setting (i.e., relatively deep layer $K=20$) to different VLLMs for generality without careful tuning. The ablations on the pruning layer $K$ demonstrate the robustness and generalizability of our proposed method. We hope the above analysis can address your concern.
>
> We appreciate your feedback and are happy to address any further suggestions or concerns you may have.
>
> [1] Growing a Twig to Accelerate Large Vision-Language Models (ICCV 2025)

---

> ### Author Response · Authors · 2025-11-27
> **Hoping for a Discussion on Our Rebuttal for Paper 5848**
>
> Dear Reviewer,
>
> I hope this message finds you well. We have carefully responded to the questions and concerns you previously raised, and we would greatly appreciate any further feedback you might be able to provide. Your insights are highly valuable to us, and we remain fully available to clarify or refine any remaining points.
>
> As the discussion deadline approaches, we would appreciate your response at your earliest convenience, as it would greatly help us move the discussion forward.
>
> Thank you sincerely for your time and effort in reviewing our paper.
>
> Best regards,
>
> The Authors

---

### Official Review · Reviewer_7jpn · 2025-11-03

**Soundness:** 3
**Presentation:** 3
**Contribution:** 3
**Rating:** 8
**Confidence:** 4

**Summary:**

This paper proposes FlashVID, a training-free inference acceleration framework for Video Large Language Models (VLLMs) that addresses the computational bottleneck of processing high volumes of visual tokens. FlashVID is a training-free acceleration framework for Video Large Language Models (VLLMs) that addresses computational bottlenecks from processing many visual tokens. Unlike existing methods that compress spatial and temporal redundancy separately, FlashVID jointly models spatiotemporal relationships through: (1) TSTM - tree-based merging that tracks tokens across frames despite spatial movement, and (2) ADTS - diversity-based selection of informative tokens per frame. Experiments on three VLLMs across five benchmarks show FlashVID retains 99.1% performance with only 10% of tokens and enables processing 10× more frames under fixed compute budgets.

**Strengths:**

- The paper identifies a clear limitation in existing methods that temporal redundancy is typically defined by fixed spatial locations, which fails to capture video dynamics where objects move, scale, and rotate. The tree-based spatiotemporal merging is an elegant solution to this problem.

- The experiments are comprehensive, covering three diverse VLLMs and five benchmarks. The results consistently show FlashVID outperforming baselines.

- The method requires no additional training, making it immediately applicable to existing VLLMs and reducing deployment barriers.

- The paper includes proper ablations (Table 4, 5, 9-12), efficiency analysis (Table 13), and extensive visualizations (Figures 5-8) that support the main claims.

**Weaknesses:**

- Limited novelty in individual components: ADTS essentially combines existing techniques ([CLS] attention + diversity-based selection via MMDP). The calibrated MMDP formulation (Algorithm 4) is relatively straightforward. The tree construction in TSTM (Algorithm 1, lines 9-16) is a simple greedy nearest-neighbor matching with thresholding. The "tree" structure emerges naturally but isn't explicitly optimized.

- The paper states that depth and breadth constraints "yielded negligible gains" (page 6, line 297-299; Table 11-12), which raises questions about whether the tree structure itself is crucial or if simple nearest-neighbor merging suffices.

- The paper combines multiple techniques: video partitioning, ADTS, TSTM, and inner-LLM pruning. Table 4-5 only ablate ADTS variants (ATS, DTS, ADTS) and \alpha, not the full pipeline. More ablation to show each contribution would make the paper more stronger.

**Questions:**

- What is the individual contribution of TSTM vs. ADTS? How much does video partitioning contribute?

- TSTM vs. simpler alternatives: Have you compared TSTM against simpler temporal merging strategies like:
  - Merging based on global feature similarity (all-pairs matching) without spatial constraints?
  - k-means clustering across frames?
  - Optical flow-guided merging?

- Given that depth/breadth constraints don't help (Table 11-12), is the tree structure essential, or is TSTM effectively performing similarity-thresholded pairwise merging?

- How does FlashVID perform on video generation or video editing tasks where spatial precision is critical? Can it handle multi-view videos or videos with significant camera motion?

---

> ### Author Response · Authors · 2025-11-21
> **Response to Weakness 1 (Limited novelty of FlashVID)**
>
> > **Weakness 1**: Limited novelty in individual components: ADTS essentially combines existing techniques ([CLS] attention + diversity-based selection via MMDP). The calibrated MMDP formulation (Algorithm 4) is relatively straightforward. The tree construction in TSTM (Algorithm 1, lines 9-16) is a simple greedy nearest-neighbor matching with thresholding. The "tree" structure emerges naturally but isn't explicitly optimized.
>
> Thank you for your comments.
>
> It is worth noting that FlashVID is a **simple yet effective** token compression framework for Video Large Language Models (VLLMs). However, simplicity doesn't mean limited novelty.
>
> Existing methods (e.g., PruneVID) fail to model the dynamic and evolving nature of video via Temporal Token Merging (TTM) strategy under rigid spatial constraints. To address this issue,  we introduce FlashVID, an efficient and effective framework for VLLMs that selects the representative visual tokens in each frame via Attention and Diversity-based Token Selection (ADTS) and further eliminates video redundancy by the Tree-based Spatiotemporal Token Merging (TSTM) mechanism, which enables fine-grained spatiotemporal compression while maintaining informative visual information.
>
> The following table (extracted from **Table 1-2 and Table 6**) compares FlashVID with FastVID on different VLLMs and diverse video understanding benchmarks. The results show that FlashVID significantly outperforms the current state-of-the-art method (i.e., FastVID), demonstrating the robustness and generalizability of our method.
>
> |Method|Retention Ratio (%)|VideoMME|EgoSchema|LongVideoBench|MVBench|Rel. Acc (%)|
> |---|---|---|---|---|---|---|
> |LLaVA-OneVision|100|58.5|60.3|56.6|58.3|100.0|
> |FastVID|10|57.2|58.7|55.7|57.0|97.8|
> |FlashVID|10|**57.8**|**60.0**|**56.5**|**57.4**|**99.1**|
> |LLaVA-Video|100|64.2|57.3|59.5|61.9|100.0|
> |FastVID|10|59.8|52.4|56.9|59.3|94.1|
> |FlashVID|10|**60.9**|**54.9**|**57.7**|**59.3**|**95.9**|
> |Qwen2.5-VL|100|61.3|58.3|58.9|68.0|100.0|
> |FastVID|10|56.3|55.6|55.4|62.3|93.2|
> |FlashVID|10|**57.3**|**55.9**|**57.1**|**65.5**|**95.6**|
>
> In addition, the table below (extracted from **Table 12**) compares TTM and TSTM in our FlashVID. The results indicate that TSTM achieves better spatiotemporal redundancy compression than the traditional TTM strategy.
>
> |Method|VideoMME|EgoSchema|LongVideoBench|MVBench|Rel. Acc (%)|
> |---|---|---|---|---|---|
> |TTM|57.3|**60.1**|56.0|57.2|98.6|
> |TSTM|**57.8**|60.0|**56.5**|**57.4**|**99.1**|
>
> More importantly, FlashVID can serve as a training-free and plug-and-play module, enabling VLLMs to process more video frames while maintaining overall computational budget. When integrating into Qwen2.5-VL, FlashVID enables **10x** video frames input, which improves **8.6%** relative performance within the same computational. The part of the results are presented in the following table (extracted from **Table 3**).
>
> |Method|#Frames|Retention Ratio (%)|VideoMME|EgoSchema|LongVideoBench|MLVU|Rel. Acc (%)|
> |---|---|---|---|---|---|---|---|
> |Qwen2.5-VL|16 (1x)|100|57.0|55.6|56.9|40.6|100.0|
> |FastVID|160 (10x)|10|61.9|59.1|58.0|43.8|105.9|
> |FlashVID|160 (10x)|10|**62.4**|**59.5**|**58.9**|**47.5**|**108.6**|
>
> In summary, our FlashVID is a simple yet effective acceleration framework for VLLMs. Despite its simplicity, it largely outperforms previous state-of-the-art methods (see **Table 1-3, Table 6-8**) across all settings (i.e., different VLLMs, evaluation benchmarks, and retention ratios), showcasing the superior performance and generalizability of our proposed method. Please note that simplicity doesn't mean limited novelty. We hope our work can inspire more in-depth research and further propose more complex and effective approaches.

---

> ### Author Response · Authors · 2025-11-21
> **Response to Weakness 2 and Questions 3 (Tree structure necessity)**
>
> > **Weakness 2**: The paper states that depth and breadth constraints "yielded negligible gains" (page 6, line 297-299; Table 11-12), which raises questions about whether the tree structure itself is crucial or if simple nearest-neighbor merging suffices.
> >
> > **Question 3**: Given that depth/breadth constraints don't help (Table 11-12), is the tree structure essential, or is TSTM effectively performing similarity-thresholded pairwise merging?
>
> Thank you for your insightful comments.
>
> Although the tree depth and breadth constraints don't bring performance gains (see **Table 11 and Table 12**), tree structure remains essential to achieve fine-grained spatiotemporal redundancy compression.
>
> In this work, we observed the dynamic nature of video where the most semantically similar visual elements are likely to experience changes in spatial position, scale, orientation, and other attributes over time. Consequently, the most correlated visual features in adjacent frames may not reside at the same spatial location. Existing methods (e.g., PruneVID) often compress temporal and spatial redundancy independently, neglecting the intrinsic spatiotemporal relationships in videos. In addition, the Temporal Token Merging (TTM) they employed is characterized by its **linear** structure (see Figure 1(a)) that measures temporal redundancy as visual feature consistency at **fixed** spatial position, resulting in merging of less correlated tokens.
>
> Based on the above observations, we introduce the Tree-based Spatiotemporal Token Merging (TSTM) module, which is designed to **jointly model the spatial and temporal redundancy** via spatiotemporal redundancy trees, without rigid spatial constraints. This tree structure in TSTM enables flexible capturing of the fine-grained spatiotemporal relationships, while the linear-structure TTM can't achieve this. A conceptual comparison between TTM and TSTM can be found in **Figure 3**, additional visualizations of TSTM can be found in **Figure 5**.
>
> The following table (extracted from **Table 12**)compares TTM and TSTM in our FlashVID. The results indicate that TSTM achieves better spatiotemporal redundancy compression than linear-structure TTM strategy.
>
> |Method|VideoMME|EgoSchema|LongVideoBench|MVBench|Rel. Acc (%)|
> |---|---|---|---|---|---|
> |TTM|57.3|**60.1**|56.0|57.2|98.6|
> |TSTM|**57.8**|60.0|**56.5**|**57.4**|**99.1**|
>
> As illustrated in **Table 11** and **Table 12**, we explicitly evaluated tree depth and breadth constraints to explore whether they can refine merging. However, results indicate that the merging threshold $T_\tau$ is sufficient to control merging strength and preserve spatiotemporal locality. Please note that it doesn't undermine the role of the tree structure; rather, it shows that the redundancy trees are **inherently adaptive** to video dynamics without needing extra parameters (i.e., tree depth and breadth constraints).
>
> To summarize, the tree structure is essential for capturing fine-grained spatiotemporal relationships in videos, and its effectiveness and generalizability are validated by the superior performance (see **Table 1-3 and Table 6-8**) on **three** representative VLLMs (i.e., LLaVA-OneVision, LLaVA-Video, and Qwen2.5-VL) on **five** widely-used video understanding benchmarks (i.e., VideoMME, EgoSchema, LongVideoBench, MVBench, MLVU).

---

> ### Author Response · Authors · 2025-11-21
> **Response to Weakness 3 and Question 1 (More ablation studies)**
>
> > **Weakness 3**: The paper combines multiple techniques: video partitioning, ADTS, TSTM, and inner-LLM pruning. Table 4-5 only ablate ADTS variants (ATS, DTS, ADTS) and $\alpha$, not the full pipeline. More ablation to show each contribution would make the paper more stronger.
> >
> > **Question 1**: What is the individual contribution of TSTM vs. ADTS? How much does video partitioning contribute?
>
> We appreciate your insightful suggestion that comprehensive ablation studies on our FlashVID would strengthen our work.
>
> As illustrated in **Section 4.3 and Appendix A.3**, we've conducted extensive ablation studies on our method. In **Table 4**, we provided ablation studies on frame-wise token selection variants. In **Table 5**, we presented ablation studies on the token distribution factor $\alpha$ in **Table 5**, where $\alpha=0$ and $\alpha=1$ denote TSTM-only and ADTS-only. Other ablations are presented in Appendix A.3. We ablated the merging threshold $T_\tau$ in TSTM (see **Table 9**), which controls the merging strength that a lower $T_\tau$ incurs more aggressive compression. Moreover, we ablated the expansion factor $f_e$ (see **Table 10**), in which a large $f_e$ may lead to computational inefficiency, while a low $f_e$ may lose critical information under extreme compression.
>
> To address the reviewer's concern, we conduct additional ablation studies on our method. By default, the ablations are conducted on LLaVA-OneVision under a 10% retention ratio.
>
> **First**, we present comprehensive module ablations ("w/o ADTS" and "w/o TSTM" were already done in **Table 5**). The experimental results show that ADTS-only ($\alpha=1$) works better than TSTM-only ($\alpha=0$), demonstrating that ADTS can select the most representative visual tokens, and TSTM-only may lead to a suboptimal spatiotemporal redundancy compression without ADTS. However, the optimal performance is achieved at $\alpha=0.7$ (see **Table 5**), implying that a balanced integration of these two modules (i.e., ADTS and TSTM) is necessary to maintain the model performance. The module with the least performance contribution should be video partitioning, which is built upon DySeg[1]. Notably, these new results are added in **Table 13 in Appendix A.3** in the revision.
>
> |Method|VideoMME|EgoSchema|LongVideoBench|MVBench|Rel. Acc (%)|
> |---|---|---|---|---|---|
> |FlashVID|57.8|60.0|**56.5**|57.4|**99.1**|
> |w/o ADTS|56.7|60.2|55.3|55.6|97.4|
> |w/o TSTM|56.9|60.1|55.6|**57.6**|98.5|
> |w/o Video Partition|**58.1**|60.1|54.8|57.4|98.6|
> |w/o Inner-LLM Pruning|56.5|**60.4**|55.1|56.5|97.8|
>
> **Second**, we present more experimental results on inner-LLM pruning. The results showcase that FlashVID only benefits from Inner-LLM Pruning strategy at extreme compression (i.e., 10% retention ratio). At higher retention ratios (15–25%), the performance is nearly identical or even slightly better **without** inner-LLM pruning. Thus, the main performance gain arises from our **before-LLM compression modules (ADTS + TSTM)**, not from simply leveraging inner-LLM pruning. The new results are added in **Table 15 in Appendix A.3** in the revision.
>
> |Method|Retention Ratio (%)|VideoMME|Egoschema|LongVideoBench|MVBench|Rel. Acc. (%)|
> |---|---|---|---|---|---|---|
> |LLaVA-OneVision|100|58.5|60.3|56.6|58.3|100.0|
> |FlashVID|25|59.2|60.4|56.8|58.0|100.3|
> |w/o inner-LLM pruning|25|58.2|60.1|58.3|58.3|100.5|
> |FlashVID|20|58.2|60.1|58.5|58.2|100.5|
> |w/o inner-LLM pruning|20|58.7|60.2|56.2|57.6|99.7|
> |FlashVID|15|58.2|60.4|57.5|57.9|100.2|
> |w/o inner-LLM pruning|15|58.2|60.3|58.0|57.3|100.2|
> |FlashVID|10|57.8|60.0|56.5|57.4|99.1|
> |w/o inner-LLM pruning|10|56.5|60.4|55.1|56.5|97.8|
>
> **Third**, we ablate the two calibration terms in ADTS. The "Event Relevance" outperforms the "[CLS] attention" calibration term, and the optimal performance is achieved when combining these two calibration terms. The new results are added in **Table 14 in Appendix A.3** in our revision.
>
> |Method|VideoMME|EgoSchema|LongVideoBench|MVBench|Rel. Acc.|
> |---|---|---|---|---|---|
> |DTS|55.7|**60.3**|55.3|55.5|97.1|
> |w/o Event Relevance|56.0|60.2|55.1|56.8|97.6|
> |w/o [CLS] attention|57.3|59.7|55.7|57.3|98.5|
> |ADTS|**57.8**|60.0|**56.5**|**57.4**|**99.1**|
>
> In conclusion, we already provided extensive ablation studies in our paper (see **Section 4.3** and **Appendix A.3**). Here, we further ablate the modules in FlashVID and the calibration terms in ADTS. We hope the above results can address the reviewer's concern regarding the limited ablations.
>
> We sincerely appreciate your feedback and are happy to address any further suggestions or concerns you may have.
>
> [1] FastVID: Dynamic Density Pruning for Fast Video Large Language Models (NeurIPS 2025)

---

> ### Author Response · Authors · 2025-11-21
> **Response to Question 2 (Compare TSTM to simpler temporal merging strategies)**
>
> > **Question 2**: TSTM vs. simpler alternatives: Have you compared TSTM against simpler temporal merging strategies like:
> >
> > - Merging based on global feature similarity (all-pairs matching) without spatial constraints?
> > - k-means clustering across frames?
> > - Optical flow-guided merging?
>
> We appreciate your insightful suggestions.
>
> Notably, we've conducted ablation studies on tree depth and breadth constraints of our Tree-based Spatiotemporal Token Merging (TSTM) module (see **Table 11, Table 12**). We found that the final performance don't benefit from depth and breadth constraints; thus, no such constraints are applied in our TSTM. Hence, merging based on global feature similarity without spatial constraints (i.e., no tree breadth constraint) is employed in our method.
>
> To address your concern, we compare our TSTM with K-Means clustering across frames and optical flow-guided merging strategies, shown in the table below. We conduct ablations on LLaVA-OneVision at a 10% retention ratio.
>
> |Method|VideoMME|EgoSchema|LongVideoBench|MVBench|Rel. Acc (%)|
> |---|---|---|---|---|---|
> |LLaVA-OneVision|58.5|60.3|56.6|58.3|100.0|
> |K-Means Clustering|57.6|**60.1**|55.7|57.2|98.7|
> |Optical Flow-guided Merging|57.5|60.0|56.0|56.8|98.5|
> |TSTM|**57.8**|60.0|**56.5**|**57.4**|**99.1**|
>
> The above experimental results demonstrate that TSTM outperforms the other simpler strategies (i.e., K-Means clustering and optical flow-guided merging).
>
> Notably, a significant limitation of K-Means clustering across frames lies in its computational inefficiency, which is **5x** slower than our TSTM. Moreover, optical flow algorithms operate on pixel space and are typically based on some fundamental assumptions (e.g., the time interval between adjacent frames is small). Therefore, it may not be applicable to those in which the adjacent sampled frames span a long time interval. In practice, we employ a dual TV-L1 optical flow algorithm to guide token merging.
>
> To summarize, our FlashVID doesn't employ the tree breadth constraint (i.e., merging based on global feature similarity without spatial constraint). When comparing to other simpler strategies (i.e., K-Means clustering across frames and optical flow-guided merging), our TSTM outperforms these two methods in efficiency and performance.
>
> We hope the above results can address your concern, and we are happy to address any further suggestions or concerns you may have.

---

> ### Author Response · Authors · 2025-11-21
> **Response to Question 4 (Application on video generation tasks)**
>
> > **Question 4**: How does FlashVID perform on video generation or video editing tasks where spatial precision is critical? Can it handle multi-view videos or videos with significant camera motion?
>
> Thank you for your questions.
>
> In this work, we propose FlashVID, a training-free token compression framework tailored to Video Large Language Models (VLLMs) towards **video understanding** tasks. Due to the significant architectural difference between video understanding and generation models, we can not directly transfer our FlashVID into video generation tasks. But adapting to video generation tasks is our future work.

---

> ### Author Response · Authors · 2025-11-27
> **Hoping for a Discussion on Our Rebuttal for Paper 5848**
>
> Dear Reviewer,
>
> I hope this message finds you well. We have carefully responded to the questions and concerns you previously raised, and we would greatly appreciate any further feedback you might be able to provide. Your insights are highly valuable to us, and we remain fully available to clarify or refine any remaining points.
>
> As the discussion deadline approaches, we would be grateful for your response at your earliest convenience, as it would greatly help us move the discussion forward.
>
> Thank you sincerely for your time and effort in reviewing our paper.
>
> Best regards,
>
> The Authors

---

### Official Review · Reviewer_JrBP · 2025-11-05

**Soundness:** 3
**Presentation:** 4
**Contribution:** 3
**Rating:** 6
**Confidence:** 3

**Summary:**

The paper proposes FlashVID, a training-free and model-agnostic token merging framework for video LLMs that jointly compresses both spatial and temporal redundancy. FlashVID introduces a two-stage pipeline: an Attention and Diversity-based Token Selection (ADTS) module that selects intra-frame tokens, and a Tree-based Spatiotemporal Token Merging (TSTM) module that merges redundant tokens across frames. Experimental results show that the proposed method can beat previous works using different backbones, and the ablation experiments are comprehensive.

**Strengths:**

1. The idea of jointly modeling spatial and temporal redundancy via a tree-based merging method is well-motivated and insightful.
2. Another component, ADTS, focusing on intra-frame redundancy, is organically combined with TSTM in the pipeline. And they are complementary.
3. Experiments are done in multiple benchmarks, with multiple backbones, compared with several previous SOTAs. The proposed model can consistently outperform them.

**Weaknesses:**

1. When comparing with other previous works, only the retention ratio and TFLOPs are used in the paper to measure the efficiency. However, token pruning is also time-consuming. Given the proposed method has many steps (eg, tree construction, similarity calculation, attention mask computation, etc), the pruning inference time complexity, and potential memory cost should also be comprehensively analyzed and compared with previous works.

2. In the paper, the author(s) claim to use a hybrid compression strategy (before-LLM + inner-LLM). However, it's unclear that given a retention ratio, how many tokens are dropped according to the proposed method (before-LLM compression), and how many tokens are dropped in inner-LLM compression. And also, how is the before-inner ratio compared to previous works? Without a detailed analysis, it is difficult to tell whether the benefit mainly comes from the proposed method or from inner-LLM compression.

**Questions:**

See weaknesses.

Besides, one minor question is: why do you introduce TSTM in sec 3.2 before ADTS in sec 3.3, tho their orders in your pipeline are opposite?

---

> ### Author Response · Authors · 2025-11-21
> **Response to Weakness 1 (Comprehensive efficiency analysis)**
>
> > **Weakness 1**: When comparing with other previous works, only the retention ratio and TFLOPs are used in the paper to measure the efficiency. However, token pruning is also time-consuming. Given the proposed method has many steps (eg, tree construction, similarity calculation, attention mask computation, etc), the pruning inference time complexity, and potential memory cost should also be comprehensively analyzed and compared with previous works.
>
> Thank you for your insightful comments.
>
> We appreciate your concern that token pruning can be a time-consuming operation. It is worth noting that FlashVID can enable Video Large Language Models (VLLMs) to achieve a satisfactory speedup with negligible performance degradation, though it consists of multiple steps (i.e., video partition, ADTS, TSTM, inner-LLM pruning).
>
> As illustrated in **Appendix A.3.1**, we've conducted an efficiency experiment on LLaVA-Video (see **Table 16** in the revision). Our FlashVID can speed up **5.3x** prefilling time and **1.9x** Time-To-First-Token (TTFT), compared to the vanilla LLaVA-Video. Following FastVID, the prefilling time contains token compression and the first LLM forward time, while the TTFT includes prefilling time and vision encoding time.
>
> To address your concern, we further conduct efficiency experiments on LLaVA-OneVision using a single NVIDIA A100 GPU, comparing to FastVID on VideoMME. As shown in the following table, we compare the prefilling time and TTFT speedups with FastVID under the same performance.
>
> |Method|Retention Ratio (%) / FLOPs (T)|GPU Memory (GB)|Vision Encoding (ms)|Compression (ms)|LLM Forward (ms)|Prefilling Time (ms)|TTFT (ms)|Rel. Acc. (%)|
> |---|---|---|---|---|---|---|---|---|
> |LLaVA-OneVision|100.0 / 113.4|30.5|785.0|-|1220.8|1220.8  (1.0x)|2005.8 (1.0x)|100.0|
> |FastVID|25.0 / 22.4|27.0|785.0|28.6|273.2|301.8 (4.0x)|1086.8 (1.8x)|99.5|
> |**FlashVID**|10.0 / 8.6|26.9|785.0|60.2|133.1|193.3 (**6.3x**)|978.3 (**2.1x**)|**99.7**|
>
> Remarkably, FlashVID preserves **99.7%** relative performance to vanilla LLaVA-OneVision under a **10%** retention ratio, while FastVID achieves a similar performance at a **25%** retention ratio. Consequently, FlashVID achieves **6.3x** prefilling and **2.1x** TTFT speedup, outperforming FastVID by a large margin.
>
> We hope the above efficiency analysis can address your concern regarding the potential token pruning inefficiency in our method.

---

> ### Author Response · Authors · 2025-11-21
> **Response to Weakness 2 (Clarification of hybrid compression strategy)**
>
> > **Weakness 2**: In the paper, the author(s) claim to use a hybrid compression strategy (before-LLM + inner-LLM). However, it's unclear that given a retention ratio, how many tokens are dropped according to the proposed method (before-LLM compression), and how many tokens are dropped in inner-LLM compression. And also, how is the before-inner ratio compared to previous works? Without a detailed analysis, it is difficult to tell whether the benefit mainly comes from the proposed method or from inner-LLM compression.
>
> We appreciate your insightful comments. A clear explanation of the hybrid compression strategy we employ and the token distribution between before-LLM and inner-LLM compression can clarify the strengths of our proposed FlashVID against previous methods.
>
> It is worth noting that we ensure fair comparisons between different acceleration frameworks under a simple token budget alignment strategy (see **Section 4.1 and Appendix C.2**). In addition, our further ablation studies (see subsequent content) on inner-LLM compression demonstrate that the main performance gains stem from our proposed Attention and Diversity-based Token Selection (ADTS) and Tree-based Spatiotemporal Token Merging (TSTM) modules.
>
> ### How are visual token numbers determined at each stage?
>
> As illustrated in **Section 4.1 and Appendix C.3**, we ensure a fair comparison by aligning the average number of visual tokens processed by each Transformer layer in the LLM, following TwigVLM[1] and DyTok[2]. To clarify, we provide a detailed explanation. Notably, the following content has been added in **Appendix C.3** in the revision.
>
> Let $\bar{R}$ be the average retained visual tokens per Transformer layer, $M$ be the number of tokens entering the LLM (after before-LLM compression), $K$ be the pruning layer index, $L$ be the number of Transformer layers in LLM, and $R$ be the number of retained visual tokens (after inner-LLM pruning). Then we have the following equation.
> $$
> \bar{R}L=MK+R(L-K).\tag{1}
> $$
>
> Solving for $R$:
>
> $$
> R=\frac{\bar{R}L-MK}{L-K}.\tag{2}
> $$
>
> FlashVID introduces a hyperparameter $f_e$, an expansion factor such that $M=f_e\bar{R}$ ; thus, we have:
>
> $$
> R=\frac{\bar{R}(L-f_eK)}{L-K}. \tag{3}
> $$
>
> And the inner-LLM pruning ratio $r$ becomes:
>
> $$
> r=\frac{R}{M}=\frac{L-f_eK}{f_e(L-K)}. \tag{4}
> $$
>
> **Example**: with $L = 28$, $f_e = 1.25$, $K = 20$, we obtain $r = 0.3$.
>
> To make the distribution of before-LLM vs inner-LLM compression explicit, we list the detailed token counts for LLaVA-OneVision under a 10% retention ratio as an example.
>
> |Method|TFLOPs|Before-LLM|Inner-LLM|
> |---|---|---|---|
> |LLaVA-OneVision|48.8|6272|6272|
> |FastV|4.7|6272|193|
> |VisionZip|4.2|627|627|
> |FastVID|4.2|627|627|
> |FlashVID|4.2|784|235|
>
> The above table clearly shows how many tokens are retained at each compression stage. It is worth noting that the computational complexity are similar between different frameworks, thereby ensuring a fair comparison.
>
> ### Does our improvement mainly come from inner-LLM compression?
>
> Here, we would like to clarify that the performance mainly benefits from our proposed ADTS and TSTM modules, instead of inner-LLM compression. To address your concern, we conduct ablation studies on inner-LLM compression, shown in the following table. Notably, we have added these results in **Appendix A.3** in the revision (see **Table 15**).
>
> |Method|Retention Ratio (%)|VideoMME|Egoschema|LongVideoBench|MVBench|Rel. Acc. (%)|
> |---|---|---|---|---|---|---|
> |LLaVA-OneVision|100|58.5|60.3|56.6|58.3|100.0|
> |FlashVID|25|59.2|60.4|56.8|58.0|100.3|
> |w/o inner-LLM compression|25|58.2|60.1|58.3|58.3|100.5|
> |FlashVID|20|58.2|60.1|58.5|58.2|100.5|
> |w/o inner-LLM compression|20|58.7|60.2|56.2|57.6|100.5|
> |FlashVID|15|58.2|60.4|57.5|57.9|100.2|
> |w/o inner-LLM compression|15|58.2|60.3|58.0|57.3|100.2|
> |FlashVID|10|57.8|60.0|56.5|57.4|99.1|
> |w/o inner-LLM compression|10|56.5|60.4|55.1|56.5|97.8|
>
> The above experimental results demonstrate that FlashVID only benefits from inner-LLM compression in the extreme situation (i.e., 10% retention ratio). At higher retention ratios (15–25%), the performance is nearly identical or even slightly better without inner-LLM compression.
>
> In summary, our ablations disentangle the contribution of inner-LLM compression, from which the main performance gains arise from our proposed before-LLM compression modules (i.e., ADTS and TSTM), rather than simply leveraging inner-LLM compression.
>
> [1] TwigVLM: Growing a Twig to Accelerate Large Vision-Language Models (ICCV 2025)
>
> [2] Less Is More, but Where? Dynamic Token Compression via LLM-Guided Keyframe Prior (NeurIPS 2025)

---

> ### Author Response · Authors · 2025-11-21
> **Response to Question 1 (Section ordering mismatch)**
>
> > **Question 1**: One minor question is: why do you introduce TSTM in sec 3.2 before ADTS in sec 3.3, tho their orders in your pipeline are opposite?
>
> Thank you for pointing this out.
>
> Although aligning the section order with the pipeline order could reduce potential confusion, we intentionally organized in this way because the Tree-based Spatiotemporal Token Merging (TSTM) mechanism is our key contribution. It enables capturing fine-grained spatiotemporal relationships in videos by jointly modeling spatial and temporal redundancy. In addition, the synergistic Attention and Diversity-based Token Selection (ADTS) module is proposed to filter the most informative visual tokens in each frame, preventing excessively redundant information input to TSTM and enabling better spatiotemporal redundancy compression. To avoid potential confusion, we explicitly mentioned the actual pipeline of our FlashVID in **Section 1 (line 103-105)** and **Section 3 (line 333-336)**.
>
> In summary, presenting TSTM first helps readers grasp the central concept before delving into the synergistic ADTS module. To further prevent ambiguity, we explicitly specify our actual pipeline in **Section 3.1** in the revision. We hope the above analysis could address your concern regarding the section ordering mismatch.
>
> We appreciate your feedback and are happy to address any further suggestions or concerns you may have.

---

> ### Author Response · Authors · 2025-11-27
> **Hoping for a Discussion on Our Rebuttal for Paper 5848**
>
> Dear Reviewer,
>
> I hope this message finds you well. We have carefully addressed the questions and concerns you previously raised, and we would greatly appreciate any further feedback you might be able to provide. Your insights are highly valuable to us, and we remain fully available to clarify or refine any remaining points.
>
> As the discussion deadline is approaching, if possible, we would be grateful for your response at your earliest convenience, as it would greatly help us move the discussion forward.
>
> Thank you sincerely for your time and effort in reviewing our paper.
>
> Best regards,
>
> The Authors

---

### Comment · Area_Chair_pqqX · 2025-11-26
**Author-Reviewer-AC Discussion (DDL: 12/3 9PM UTC)**

Dear Reviewers,

Thank you once again for your service to ICLR 2026. Now that the authors have submitted their rebuttal, I kindly ask you to take the following steps (if you have not done so already):

- Read the authors’ response and other reviews.
- Consider whether the rebuttal and additional comments affect your assessment of the paper.
- Engage in interactive discussion with the authors -- **Note the Author-Reviewer-AC discussion period ends on 12/3 9PM UTC**. You are recommended to keep active before that deadline. If you have more concerns/questions (e.g., requesting clarifications, new results), it is recommended to post your request asap, so that the authors have enough time to address them.

The current reviews for this paper are **mixed (scores: 6/8/4/4)**. Your further contributions are essential for forming a well-informed final decision.

I am happy to join and support the discussions between you and the authors. Please feel free to share your thoughts and participate actively in the discussion. Thanks!

Best regards,

AC

---

### Author Response · Authors · 2025-12-01
**Summary of Rebuttal**

Dear Area Chairs (ACs),

We sincerely appreciate your time and effort in reviewing our paper.

During the discussion period, we made every effort to address the concerns raised by reviewers. So far, we've received positive feedback from Reviewer nJgS that our rebuttal resolved his/her concerns. Consequently, he/she is willing to raise the score to **6** in support of the recommendation for our submission.

In addition, we would like to specify that our rebuttal to the Reviewer nJgS is based on the **updated** review on Nov. 19. (see **revisions of "Official Review of Submission5848 by Reviewer nJgS**"). This is because **the initial review from Reviewer nJgS does not match our paper**, for which we inquired with Reviewer nJgS on **Nov. 19**, and we subsequently received the correct review on the same day. (see "**Inquiry for the Mismatched Review for the Submission Paper 5848**")

Before the system reset, our rebuttal and revisions had resulted in a **score improvement from (6, 8, 4, 4) to (6, 8, 4, 6)**.

|Reviewer|Initial Rating|Final Rating|Confidence|Feedback|
|---|---|---|---|---|
|JrBP|6|6|3|None|
|7jpn|8|8|4|None|
|tpdQ|4|4|5|None|
|nJgS|4|**6** (4 -> 6)|4|Positive|

**Note**: 'None' denotes that we have not received feedback from the reviewer.

Best regards,

The authors

---

> ### Author Response · Authors · 2025-12-01
> **Summary of Rebuttal (Part 1/4): Reviewer JrBP (Initial Rating: 6, Confidence: 3, Final Rating: 6)**
>
> Reviewer JrBP praised FlashVID as **well-motivated, well-designed, and performant**, but raised main concerns regarding the efficiency of FlashVID, the unclear hybrid compression strategy, and the section ordering mismatch.
>
> > **Concern 1 (Efficiency)**: Require a comprehensive efficiency analysis
>
> - **Solution**: As required, we presented detailed efficiency analyses on LLaVA-Video (see **Table 16 in Appendix A.3.1**) and LLaVA-OneVision (see **"Response to Weakness 1 (Comprehensive efficiency analysis)"**) compared to the previous state-of-the-art method (i.e., FastVID).
>
> - **Conclusion**: FlashVID enables VLLMs to achieve a satisfactory speedup with negligible performance degradation, despite including multiple pruning steps.
>
> > **Concern 2 (Hybrid compression)**: Require clarification of the hybrid compression strategy
>
> - **Solution**: As required, we provided the detailed token budget alignment strategy in **Appendix C.3** in the revision. Moreover, we conducted ablation studies on inner-LLM compression in **Appendix A.3** in the revision, in which the results show that FlashVID only benefits from inner-LLM compression at extreme compression (i.e., 10% retention ratio).
>
> - **Conclusion**: We clarified the detailed hybrid compression strategy and demonstrated that the main performance gains stem from our proposed before-LLM compression modules (i.e., ADTS and TSTM).
>
> > **Concern 3 (Writing)**: Require clarification of section ordering mismatch
>
> - **Solution**: As required, we clarified that we intentionally organized in this way because the Tree-based Spatiotemporal Token Merging (TSTM) mechanism is our key contribution. To avoid potential confusion, we explicitly mentioned the actual pipeline of our FlashVID in **Section 1 (line 103-105) and Section 3 (line 333-336)**.
>
> - **Conclusion**: We specified the reason for the section ordering mismatch and emphasized the actual pipeline of FlashVID in the main text to avoid ambiguity.
>
> In summary, we have addressed the concerns raised by Reviewer JrBP.

---

> ### Author Response · Authors · 2025-12-01
> **Summary of Rebuttal (Part 2/4): Reviewer 7jpn (Initial Rating: 8, Confidence: 4, Final Rating: 8)**
>
> Reviewer 7jpn praised FlashVID as **an elegant solution to sptaiotemporal redundancy compression, performant, training-free, and plug-and-play**, but raised main concerns regarding the limited novelty of FlashVID, the necessity of tree structure in TSTM, incomplete ablation studies, and comparison between TSTM and simpler temporal merging strategies.
>
>
> > **Concern 1 (Novelty)**: Question the novelty of FlashVID
>
> - **Solution**: As required, we clarified that FlashVID is a **simple yet effective** method. Despite its simplicity, it outperforms previous state-of-the-art methods (e.g., VisionZip and FastVID) across all settings (see **Table 1-3, Table 6-8**). Moreover, we specified that our main contribution is that we identify that existing methods fail to capture video dynamics, and we proposed a straightforward method (i.e., TSTM) to capture fine-grained spatiotemporal relationships.
> - **Conclusion**: FlashVID is a **simple yet effective** acceleration framework for VLLMs. However, **simplicity doesn't mean limited novelty**.
>
> > **Concern 2 (Tree structure)**: Question the necessity of tree structure
>
> - **Solution**: As required, we presented a detailed analysis of the tree structure necessity in FlashVID (see "**Response to Weakness 2 and Questions 3 (Tree structure necessity)**").
> - **Conclusion**: Tree structure is essential to achieve a fine-grained spatiotemporal redundancy compression.
>
> > **Concern 3 (Ablations)**: Require more ablations
>
> - **Solution**: As required, we complemented ablation studies on modules, inner-LLM pruning, and calibration terms in ADTS (see "**Response to Weakness 3 and Question 1 (More ablation studies)**").
> - **Conclusion**: Combined with the ablation studies in **Section 4.3 and Appendix A.3** (see **Table 4-5, Table 9-14** in the revision), we presented comprehensive ablation studies on FlashVID.
>
> > **Concern 4 (Comparisons)**: Require comparisons to simpler temporal merging strategies
>
> - **Solution**: As required, we compared TSTM with simpler temporal merging strategies (i.e., K-Means clustering, optical flow-guided merging) (see "**Response to Question 2 (Compare TSTM to simpler temporal merging strategies)**").
> - **Conclusion**: TSTM outperforms all simpler alternative methods in efficiency and performance.
>
> In summary, we have addressed the concerns raised by Reviewer 7jpn.

---

> ### Author Response · Authors · 2025-12-01
> **Summary of Rebuttal (Part 3/4): Reviewer tpdQ (Initial Rating: 4, Confidence: 5, Final Rating: 4)**
>
> Reviewer tpdQ praised FlashVID as **training-free, plug-and-play and well-motivated**, but raised main concerns regarding the sensitivity of hyperparameters in our method, improper assumptions of the "less is more" pattern, potential limitation of event relevance calibration in ADTS, and the confusing choice of $K=20$ for different VLLMs.
>
> > **Concern 1 (Sensitivity)**: Require comprehensive sensitivity analysis of hyperparameters
>
> - **Solution**: we clarified that we use the **same** set of hyperparameters in all our experiments (see **Section 4.1 and Appendix C.2**). The results (see **Table 1-3, Table 6-8**) showed FlashVID consistently outperforms previous SOTA methods (e.g., VisionZip and FastVID).
> - **Conclusion**: Despite using the same set of hyperparameters, FlashVID consistently outperforms previous methods across all settings, implying the superior robustness and generalizability of our method.
>
> > **Concern 2 ("Less is more" pattern)**: Require in-depth analysis of the "less is more" pattern
>
> - **Solution**: We clarified that the "less is more" pattern is not our focus in this work, and we updated the clarification of the "less is more" pattern in **Appendix E.4** in the revision.
> - **Conclusion**: The "Less is more" pattern is not our focus in this work. Therefore, it does not undermine the contribution of our paper. We revised the related content regarding the "less is more" pattern to avoid making reasonable but unverified assumptions.
>
> > **Concern 3 (Potential limitation of ADTS)**: Require in-depth analysis of event relevance calibration in ADTS
>
> - **Solution**: We conducted ablation studies on calibration terms in ADTS. The results show that both [CLS] attention and event relevance calibration terms bring performance gains, and the peak performance is achieved using these two calibration terms. In addition, we provided visualizations on ADTS with and without event relevance (see **Figure 10** in the revision).
> - **Conclusion**: The quantitative and qualitative results demonstrate that event relevance doesn't undermine the performance on those tasks requiring fine-grained perception.
>
> > **Concern 4 (Confusing choice of K=20)**: Require clarification of consistent choice of K=20
>
> - **Solution**: We clarified that the consistent choice of the pruning layer $K=20$ for different VLLMs is based on an insight (from TwigVLM) that the cross-modal attention weights in deep layers showcase strong visual perception capabilities. Given the LLM with $L=28$ Transformer layers, we choose a relatively deep layer $K=20$ for inner-LLM pruning without careful tuning. Moreover, we conducted ablation studies on the pruning layer $K$ (see "**Response to Question 1 (Choice of K=20 for multiple models)**").
> - **Conclusion**: The consistent choice of $K=20$ is not determined by careful tuning but built upon an insight from TwigVLM, while the performance of FlashVID is robust to different pruning layers (e.g., 18-22).
>
> In summary, we have addressed the concerns raised by Reviewer tpdQ.

---

> ### Author Response · Authors · 2025-12-01
> **Summary of Rebuttal (Part 4/4): Reviewer nJgS (Initial Rating: 4, Confidence: 4, Final Rating: 6)**
>
> Reviewer nJgS praised FlashVID as **well-motivated, training-free, and plug-and-play**, but raised main concerns regarding the computational complexity of TSTM, the sensitivity to multiple hyperparameters, and incomplete visualizations on TSTM.
>
> > **Concern 1 (Efficiency)**: Require a comprehensive efficiency analysis
>
> - **Solution**: As required, we presented detailed efficiency analyses on LLaVA-Video (see **Table 16 in Appendix A.3.1**) and LLaVA-OneVision (see **"Response to Weakness 1 (Comprehensive efficiency analysis)"**) compared to the previous state-of-the-art method (i.e., FastVID).
> - **Conclusion**: FlashVID enables VLLMs to achieve a satisfactory speedup with negligible performance degradation, despite including multiple pruning steps.
>
> > **Concern 2 (Sensitivity)**: Require comprehensive sensitivity analysis of hyperparameters
>
> - **Solution**: As required, We clarified that we use the **same** set of hyperparameters in all our experiments (see **Section 4.1 and Appendix C.2**). The results (see **Table 1-3, Table 6-8**) showed FlashVID consistently outperforms previous SOTA methods (e.g., VisionZip and FastVID).
> - **Conclusion**: Despite using the same set of hyperparameters, FlashVID consistently outperforms previous methods across all settings, implying the superior robustness and generalizability of our method.
>
> > **Concern 3 (Visualizations)**: Require more visualizations of TSTM
>
> - **Solution**:  As required, we presented more visualizations on TSTM (see **Figure 5-6**), providing an intuitive merging pattern and failure cases of TSTM.
> - **Conclusion**: The visualizations reveal an intuitive merging pattern of TSTM that merges tokens with similar semantics. Moreover, TSTM still faces challenges, such as merging tokens from different entities with similar semantics.
>
> In summary, we have addressed the concerns raised by Reviewer nJgS, resulting in a score increase from 4 to 6.

---

### Meta-Review · Area_Chair_3bgr · 2026-01-08

**Summary:**

FlashVID is a training-free, model-agnostic framework designed to accelerate Video LLMs by reducing visual token redundancy. It employs a two-stage pipeline: ADTS selects the most informative intra-frame tokens based on attention and diversity, while TSTM builds a spatiotemporal tree to merge redundant tokens across frames.

By jointly modeling spatial and temporal relationships, FlashVID retains 99.1% performance with only 10% of tokens. This efficiency allows models to process 10× more frames under the same compute budget, significantly enhancing long-video understanding without requiring additional training or fine-tuning. Below I list the pros and cons raised by the reviewers:

Pros:
* Multiple reviewers appreciated the "Tree-based" merging (TSTM)  for capturing dynamic video content (moving/rotating objects) that traditional fixed-position temporal methods fail to track.
* The framework’s ability to be a "plug-and-play" solution for existing VLLMs without expensive fine-tuning is seen as a major practical advantage, which is also praised by multiple reviewers.
* The method consistently outperforms previous SOTA models across multiple backbones (LLaVA-OneVision, Qwen2-VL) and various benchmarks is a big plus.
* The pipeline successfully combines intra-frame (ADTS) and inter-frame (TSTM) compression to handle the "token explosion" in long videos.

Cons:
* The mayor concern is the Lack of Wall-Clock Time Analysis. Reviewers noted that while TFLOPs and token counts decrease, the overhead of the pruning itself (tree construction, similarity calculations) isn't analyzed. This makes it hard to judge actual speedup.
* The second concern is the ambiguity in component contributions. It is unclear how much of the performance gain is due to FlashVID versus the "inner-LLM" pruning. More detailed ablations are needed to justify the complexity of the tree structure over simpler methods.

Other relative minor concerns
* Hyperparameter Sensitivity: The method relies on multiple empirical thresholds (α,τ,η) which may require manual tuning for different datasets, potentially undermining the "plug-and-play" claim.
* Risk of Missing Fine-Grained Details: One reviewer highlighted that the Global Average Pooling (GAP) used in ADTS might cause the model to discard small, local events that are critical for fine-grained video understanding.
* Under-Analyzed "Less is More" Phenomenon: The claim that fewer tokens lead to better performance by reducing "noise" is intriguing but lacks the quantitative or qualitative depth (e.g., attention map visualizations) to be fully persuasive.

**Reviewer Concerns:**

The authors actively participated in the rebuttal period and provided ablations for the major concerns including the speedup analysis of the proposed approach, and the ablation for separating the contribution from different components. These results are clear and convincing.

**Reviewer Scores:**

The score changes is listed above:

| Reviewer	| Initial Rating |	Final Rating |	Confidence |	Feedback |
| --|--|--|--|--|
| JrBP |	6	|  6	| 3	| None |
| 7jpn	| 8	| 8	| 4	| None |
| tpdQ	| 4	| 4	| 5	| None |
| nJgS	| 4	| 6 (4 -> 6) |	4 |	Positive |

I believe the reviewers with positive scores will remain positive after rebuttal, the concers from reviewer tpdQ has already been addressed

---

### Decision · Program_Chairs · 2026-01-26

Accept (Oral)